# VARIATIONAL LANGUAGE CONCEPTS FOR INTERPRETING PRETRAINED LANGUAGE MODELS

## ABSTRACT

Pretrained Language Models (PLMs) such as BERT and its variants have achieved remarkable success in natural language processing. To date, the interpretability of PLMs has primarily relied on the attention weights in their self-attention layers. However, these attention weights only provide word-level interpretations, failing to capture higher-level structures, and are therefore lacking in readability and intuitiveness. To address this challenge, we first provide a formal definition of *conceptual interpretation* and then propose a variational Bayesian framework, dubbed VAriational LANguage ConcEpt (VALANCE), to go beyond word-level interpretations and provide concept-level interpretations. Our theoretical analysis shows that our VALANCE finds the optimal language concepts to interpret PLM predictions. Empirical results on several real-world datasets show that our method can successfully provide conceptual interpretation for PLMs.

## 1 INTRODUCTION

Pretrained language models (PLMs) such as BERT (Devlin et al., 2018) and its variants (Lan et al., 2019; Liu et al., 2019; He et al., 2021; Portes et al., 2023) have achieved remarkable success in natural language processing. These PLMs are usually large attention-based neural networks that follow a pretrain-finetune paradigm, where models are first pretrained on large datasets and then finetuned for a specific task. As with any machine learning models, interpretability in PLMs has always been a desideratum, especially in decision-critical applications (e.g., healthcare).

To date, PLMs' interpretability has primarily relied on the attention weights in self-attention layers. However, these attention weights only provide raw word-level importance scores as interpretations. Such low-level interpretations fail to capture higher-level semantic structures, and are therefore lacking in readability, intuitiveness and stability. For example, low-level interpretations often fail to capture influence of similar words to predictions, leading to unstable or even unreasonable explanations (see Sec. 5.4 for details). In this paper, we make an attempt to go beyond word-level attention and interpret PLM predictions at the concept (topic) level. Such higher-level semantic interpretations are complementary to word-level importance scores and often more readable and intuitive.

We start by developing a comprehensive and formal definition of *conceptual interpretation* with four desirable properties: (1) multi-level structure, (2) normalization, (3) additivity, and (4) mutual information maximization. With this definition, we then propose a variational Bayesian framework, dubbed VAriational LANguage ConcEpt (VALANCE), to provide *dataset-level*, *document-level*, and *word-level* (the first property) conceptual interpretation for PLM predictions. Our theoretical analysis shows that maximizing our VALANCE's evidence lower bound is equivalent to inferring the optimal conceptual interpretation with *Properties (1-3)* while maximizing the mutual information between the inferred concepts and the observed embeddings from PLMs, i.e., *Property (4)*.

The core of our idea is to treat a PLM's contextual word embeddings (and their corresponding attention weights) as observed variables and build a probabilistic generative model to automatically infer the higher-level semantic structures (e.g., concepts or topics) from these embeddings and attention weights, thereby interpreting the PLM's predictions at the concept level. Our VALANCE is compatible with any attention-based PLMs and can work as an conceptual interpreter, which explains the PLM predictions at multi-levels with theoretical guarantees. Our contributions are as follows:

- We identify the problem of multi-level interpretations for PLM predictions, develop a formal definition of *conceptual interpretation*, and propose VALANCE as the first general method to infer such conceptual interpretation.
- Our theoretical analysis shows that learning our VALANCE is equivalent to inferring the optimal conceptual interpretation according to our definition.
- Quantitative and qualitative analysis on three real-world datasets show that VALANCE can infer meaningful language concepts to effectively and intuitively interpret PLM predictions.

## 2 RELATED WORK

**Pretrained Language Models.** Pretrained language models are large attention-based neural networks that follow a pretrain-finetune paradigm. Usually they are first pretrained on large datasets in a self-supervised manner and then finetuned for a specific downstream task. BERT (Devlin et al., 2018) is a pioneering PLM that has shown impressive performance across multifple downstream tasks. Following BERT, there have been variants, such as ALBERT (Lan et al., 2019), Distil-BERT (Sanh et al., 2019), and Tinybert (Jiao et al., 2019), that achieve performance comparable to BERT with fewer parameters. Other variants such as RoBERTa (Liu et al., 2019) and BART (Lewis et al., 2019) improve the performance using more sophisticated training schemes for the masked language modeling learning objective. More recently, there have also been BERT variants that design different self-supervised learning objectives to achieve better performance; examples include DeBERTa (He et al., 2021), ELECTRA (Clark et al., 2020), and XLNet (Yang et al., 2019). While these PLMs naturally provide attention weights for each word to intepret model predictions, such low-level interpretations fail to capture higher-level semantic structures, and are therefore lacking in readability and intuitiveness. In contrast, our VALANCE goes beyond word-level attention and interpret PLM predictions at the concept (topic) level. These higher-level semantic interpretations are complementary to word-level importance scores and tend to more readable and intuitive.

**Topic Models.** Our work is also related to topic models Blei (2012); Blei et al. (2003), which typically build upon latent Dirichlet allocation (LDA) (Blei et al., 2003). Topic models takes the (discrete) bag-of-words representations of the documents (i.e., vocabulary-length vectors that count word occurrences) as input, discover hidden topics from them during training, and infer the topic proportion vector for each document during inference (Blei et al., 2003; Blei & Lafferty, 2006; Wang et al., 2012; Chang & Blei, 2009). Besides these 'shallow' topic models, there has been recent work that employs 'deep' neural networks to learn topic models more efficiently (Card et al., 2017; Xing et al., 2017; Peinelt et al., 2020), using techniques such as amortized variational inference. There is also work that improves upon traditional topic models by either leveraging word similarity as a regularizer for topic-word distributions (Das et al., 2015; Batmanghelich et al., 2016) or including word embeddings into the generative process (Hu et al., 2012; Dieng et al., 2020; Bunk & Krestel, 2018; Duan et al., 2021).

There are also works that build topic models upon embeddings from PLMs (Grootendorst, 2020; Zhang et al., 2022; Wang et al., 2022; Zhao et al., 2020; Meng et al., 2022). However, they typically use a pipeline consisting of dimensionality reduction followed by a simple clustering algorithm; they are not end-to-end and therefore often suffer from information loss between the PLMs' embeddings and the clustering results, leading to interpretations that are not faithful to the target PLM. Moreover, they can only generate single-level concepts, e.g., document-level concepts, and fail to provide the multi-level structure of conceptual interpretations (the first property of our definition). In contrast, our method is inherently multi-level and end-to-end, models concepts across dataset, document and word levels, and produces faithful post-hoc interpretations for any prediction models based on PLM embeddings with theoretical guarantees.

## 3 METHODS

In this section, we formalize the definition of *conceptual interpretation*, and describe our proposed VALANCE for conceptual interpretation of PLMs.

## 3.1 PROBLEM SETTING AND NOTATION

We consider a corpus of $M$ documents, where the $m$'th document contains $J_m$ words, and a PLM $f(\mathcal{D}_m)$, which takes as input the document $m$ (denoted as $\mathcal{D}_m$) with $J_m$ words and outputs (1) a CLS embedding $\mathbf{c}_m \in \mathbb{R}^d$, (2) $J_m$ contextual word embeddings $\mathbf{e}_m \triangleq [\mathbf{e}_{mj}]_{j=1}^{J_m}$, and (3) the attention weights $\mathbf{a}_m^{(h)} \triangleq [a_{mj}^{(h)}]_{j=1}^{J_m}$ between each word and the last-layer CLS token, where $h$ denotes the $h$'th attention head. We denote the average attention weight over H heads as $a_{mj} = \frac{1}{H}\sum_{h=1}^{H} a_{mj}^{(h)}$ and correspondingly $\mathbf{a}_m \triangleq [a_{mj}]_{j=1}^{J_m}$ (see the PLM at the bottom of Fig. 1).

In PLMs, these last-layer CLS embeddings are used as document-level representations for downstream tasks (e.g., document classification). Furthermore, our VALANCE assumes $K$ concepts (topics) for the corpus. For document $m$, our VALANCE interpreter tries to infer a concept distribution vector $\boldsymbol{\theta}_m \in \mathbb{R}^K$ (also known as the topic proportion in topic models) for the whole document and a concept distribution vector $\boldsymbol{\phi}_{mj} = [\phi_{mjk}]_{k=1}^{K} \in \mathbb{R}^K$ for word $j$ in document $m$. In our continuous embedding space, the $k$'th concept is represented by a Gaussian distribution, $\mathcal{N}(\boldsymbol{\mu}_k, \boldsymbol{\Sigma}_k)$, of contextual word embeddings; we use shorthand $\boldsymbol{\Omega}_k = (\boldsymbol{\mu}_k, \boldsymbol{\Sigma}_k)$ for brevity. The goal is

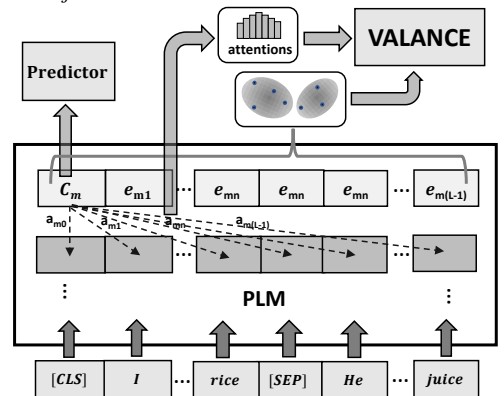

Figure 1: Overview of VALANCE framework.

to interpret PLMs' predictions *at the concept level* using the inferred document-level concept vector $\boldsymbol{\theta}_m$, word-level concept vector $\boldsymbol{\phi}_{mj}$, and the learned embedding distributions $\{\mathcal{N}(\boldsymbol{\mu}_k, \boldsymbol{\Sigma}_k)\}_{k=1}^{K}$ for each concept (see Sec. 5.4 for detailed descriptions and visualizations).

## 3.2 FORMAL DEFINITION OF LANGUAGE CONCEPTS

Below we formally define "conceptual interpretation" for PLM predictions (see notations in Sec. 3.1):

**Definition 3.1** (**Conceptual Interpretation**). Assume $K$ concepts and a dataset $\mathcal{D}$ containing $M$ documents, each with $J_m$ words ($m \in \{1, \ldots, M\}$). Conceptual interpretation for a document $m$ consists of $K$ *dataset-level* variables $\{\boldsymbol{\Omega}_k\}_{k=1}^{K}$, a *document-level* variable $\boldsymbol{\theta}_m$, and $J_m$ *word-level* variables $\{\boldsymbol{\phi}_{mj}\}_{j=1}^{J_m}$ with the following properties:

(1) **Multi-Level Structure.** Conceptual interpretation has a three-level structure:
   (a) Each *dataset-level* variable $\boldsymbol{\Omega}_k = (\boldsymbol{\mu}_k, \boldsymbol{\Sigma}_k)$ describes the $k$'th concept; $\boldsymbol{\mu}_k \in \mathbb{R}^d$ and $\boldsymbol{\Sigma}_k \in \mathbb{R}^{d \times d}$ denote the mean and covariance of the concept in the embedding space $\mathbb{R}^d$, respectively.
   (b) Each *document-level* variable $\boldsymbol{\theta}_m \in \mathbb{R}_{\geq 0}^K$ describes document $m$'s relation to the $K$ concepts.
   (c) Each *word-level* variable $\boldsymbol{\phi}_{mj} \in \mathbb{R}_{\geq 0}^K$ describes word $j$'s relation to the $K$ concepts.
(2) **Normalization.** The document- and word-level interpretations, $\boldsymbol{\theta}_m$ and $\boldsymbol{\phi}_{mj}$, are normalized:
   (a) $\sum_{k=1}^{K} \theta_{mk} = 1$ for document $m$.
   (b) $\sum_{k=1}^{K} \phi_{mjk} = 1$ for word $j$ in document $m$.
(3) **Additivity.** We can add/subtract the $k$'s concept from the contextual embeddings $\mathbf{e}_{mj}$ of word $j$ in document $m$, i.e. $\mathbf{e}_{mj} \leftarrow \mathbf{e}_{mj} \pm x_k \boldsymbol{\mu}_k$, where $x_k$ is the editing weight of the $k$'s concept.
(4) **Mutual Information Maximization.** The conceptual interpretation achieves maximum mutual information between the observed contextual embeddings $\mathbf{e}_m$ in PLMs and the document-level/word-level interpretation, $\boldsymbol{\theta}_m$ and $\boldsymbol{\phi}_{mj}$.

In Definition 3.1, Property (1) provides comprehensive three-level conceptual interpretation for PLM predictions, Property (2) ensures proper normalization in concept assignment at the document and word levels, Property (3) enables better concept editing (more details in Sec. 5.3) to modify PLM predictions, and Property (4) ensures minimal information loss when interpreting PLM predictions.

### 3.3 VARIATIONAL LANGUAGE CONCEPTS (VALANCE)

**Method Overview.** We then propose our model, VAriational LANguage ConcEpts (VALANCE), to infer the optimal conceptual interpretation described in Definition 3.1. Different from *static* word embeddings (Mikolov et al., 2013) and topic models, PLMs produce *contextual* word embeddings with continuous-value entries $[\mathbf{e}_{mj}]_{j=1}^{J_m}$ and more importantly, associate each word embedding with a continuous-value attention weight $[a_{mj}]_{j=1}^{J_m}$; therefore this brings unique challenges.

To effectively discover latent concept structures learned by PLMs at the dataset level and interpret PLM predictions at the data-instance level, our VALANCE treats both the contextual word embeddings and their associated attention weights as observations to learn a probabilistic generative model of these observations, as shown in Fig. 1. The key idea is to use the attention weights from PLMs to compute a virtual continuous count for each word, and model the contextual word embedding distributions with Gaussian mixtures. The generative process of VALANCE is as follows (we mark key connection to PLMs in blue and show the corresponding graphical model in Fig. 2):

For each document $m, 1 \leq m \leq M$,

1. Draw the document-level concept distribution vector $\boldsymbol{\theta}_m \sim \text{Dirichlet}(\boldsymbol{\alpha})$.
2. For each word $j$ $(1 \leq j \leq J_m)$,
   (a) Draw the word-level concept index $z_{mj} \sim \text{Categorical}(\boldsymbol{\theta}_m)$.
   (b) With a continuous word count $w_{mj} \in \mathbb{R}$ from the PLM's attention weights, Draw the contextual word embedding of the PLM from the corresponding Gaussian component $\mathbf{e}_{mj} \sim \mathcal{N}(\boldsymbol{\mu}_{z_{mj}}, \boldsymbol{\Sigma}_{z_{mj}})$.

Given the generative process above, discovery of latent concept structures in PLMs at the dataset level boils down to learning the parameters $\{\boldsymbol{\mu}_k, \boldsymbol{\Sigma}_k\}_{k=1}^K$ for the $K$ concepts. Intuitively the global parameters $\{\boldsymbol{\mu}_k, \boldsymbol{\Sigma}_k\}_{k=1}^K$ are shared across different documents, and they define a mixture of $K$ Gaussian distributions. Each Gaussian distribution describes a 'cluster' of words and their contextual word embeddings.

Similarly, interpretations of PLM predictions at the data-instance level is equivalent to inferring the latent variables, i.e., document-level concept distribution vectors $\boldsymbol{\theta}_m$ and word-level concept indices $z_{mj}$. Below we highlight several important aspects of our VALANCE designs.

**Attention Weights as *Continuous* Word Counts.** Different from typical topic models (Blei et al., 2003; Blei, 2012) and word embeddings (Mikolov et al., 2013) that can only handle *discrete* word counts, our VALANCE can handle *continuous* (virtual) word counts; this better aligns with continuous attention weights in PLMs. Specifically, we denote as $w_{mj} \in \mathbb{R}_{\geq 0}$ the (non-negative real-valued) *continuous word count* for the $j$'th word in document $m$. We explore three schemes of computing $w_{mj}$:

* **Identical Weights:** Use identical weights for different words, i.e., $w_{mj} = 1, \forall m, j$. This is equivalent to typical discrete word counts.
* **Attention-Based Weights with Fixed Length:** Use $w_{mj} = J' a_{mj}$, where $J'$ is a fixed sequence length shared across all documents.
* **Attention-Based Weights with Variable Length:** Use $w_{mj} = J_m a_{mj} / \sum_{i=1}^{J_m} a_{mi}$, where $J_m$ is true sequence length without padding. Note that in practice, $\sum_{i=1}^{J_m} a_{mi} \neq 1$ due to padding tokens in PLMs.

**Contextual Continuous Word Representations.** Note that different from topic models (Blei et al., 2003) and typical word embeddings (Mikolov et al., 2013; Dieng et al., 2020) where word representations are *static*, word representations in PLMs are *contextual*; specifically, the same word can have different embeddings in different documents (contexts). For example, the word 'soft' can appear as the $j_1$'th word in document $m_1$ and as the $j_2$'th word in document $m_2$, and therefore have two different embeddings (i.e., $\mathbf{e}_{m_1 j_1} \neq \mathbf{e}_{m_2 j_2}$).

Correspondingly, in our VALANCE, we do not constrain the same word to have a static embedding; instead we assume that a word embedding is drawn from a Gaussian distribution corresponding to its latent topic. It is also worth noting that word representations in VALANCE is continuous, which is different from typical topic models (Blei et al., 2003) based on (discrete) bag-of-words representations.

## 3.4 OBJECTIVE FUNCTION

Below we discuss the inference and learning procedure for VALANCE. We start by introducing the *inference* of document-level and word-level concepts (i.e., $z_{mj}$ and $\boldsymbol{\theta}_m$) given the global concept parameters (i.e., $\{(\boldsymbol{\mu}_k, \boldsymbol{\Sigma}_k)\}_{k=1}^K$), and then introduce the *learning* of these global concept parameters.

### 3.4.1 INFERENCE

**Inferring Document-Level and Word-Level Concepts.**
We formulate the problem of interpreting PLM predictions at the concept level as inferring document-level and

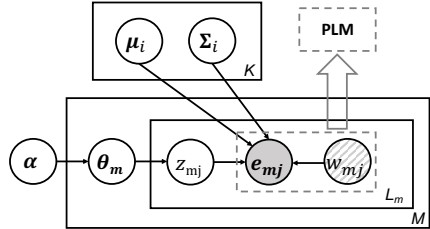

Figure 2: Graphical model of our VALANCE. The *striped* circle represents *continuous* word counts.

word-level concepts. Specifically, given global concept parameters $\{(\boldsymbol{\mu}_k, \boldsymbol{\Sigma}_k)\}_{k=1}^K$, the *contextual* word embeddings $\mathbf{e}_m \triangleq [\mathbf{e}_{mj}]_{j=1}^{J_m}$, and the associated attention weights $\mathbf{a}_m \triangleq [a_{mj}]_{j=1}^{J_m}$, a PLM produces for each document $m$, our VALANCE infers the posterior distribution of the document-level concept vector $\boldsymbol{\theta}_m$, i.e., $p(\boldsymbol{\theta}_m|\mathbf{e}_m, \mathbf{a}_m, \{(\boldsymbol{\mu}_k, \boldsymbol{\Sigma}_k)\}_{k=1}^K)$, and the posterior distribution of the word-level concept index $z_{mj}$, i.e., $p(z_{mj}|\mathbf{e}_m, \mathbf{a}_m, \{(\boldsymbol{\mu}_k, \boldsymbol{\Sigma}_k)\}_{k=1}^K)$.

**Variational Distributions.** These posterior distributions are intractable; we therefore resort to variational inference (Jordan et al., 1998; Blei et al., 2003) and use variational distributions $q(\boldsymbol{\theta}_m|\boldsymbol{\gamma}_m)$ and $q(z_{mj}|\boldsymbol{\phi}_{mj})$ to approximate them. Here $\boldsymbol{\gamma}_m \in \mathbb{R}^K$ and $\boldsymbol{\phi}_{mj} \triangleq [\phi_{mjk}]_{k=1}^K \in \mathbb{R}^K$ are variational parameters to be estimated during inference. This leads to the following joint variational distribution:

$$q(\boldsymbol{\theta}_m, \{\mathbf{z}_{mj}\}_{j=1}^{J_m}|\boldsymbol{\gamma}_m, \{\boldsymbol{\phi}_{mj}\}_{j=1}^{J_m}) = q(\boldsymbol{\theta}_m|\boldsymbol{\gamma}_m) \cdot \prod_{j=1}^{J_m} q(z_{mj}|\boldsymbol{\phi}_{mj}) \tag{1}$$

**Evidence Lower Bound.** For each document $m$, finding the optimal variational distributions is then equivalent to maximizing the following evidence lower bound (ELBO):

$$\mathcal{L}(\boldsymbol{\gamma}_m, \{\boldsymbol{\phi}_{mj}\}_{j=1}^{J_m}; \boldsymbol{\alpha}, \{(\boldsymbol{\mu}_k, \boldsymbol{\Sigma}_k)\}_{k=1}^K) = \mathbb{E}_q[\log p(\boldsymbol{\theta}_m|\boldsymbol{\alpha})] + \sum_{j=1}^{J_m} \mathbb{E}_q[\log p(z_{mj}|\boldsymbol{\theta}_m)]$$

$$+ \sum_{j=1}^{J_m} \mathbb{E}_q[\log p(\mathbf{e}_{mj}|z_{mj}, \boldsymbol{\mu}_{z_{mj}}, \boldsymbol{\Sigma}_{z_{mj}})] - \mathbb{E}_q[\log q(\boldsymbol{\theta}_m)] - \sum_{j=1}^{J_m} \mathbb{E}_q[\log q(z_{mj})], \tag{2}$$

where the expectation is taken over the joint variational distribution in Eqn. 1.

**Likelihood with *Continuous* Word Counts.** One key difference between VALANCE and typical topic models (Blei et al., 2003; Blei, 2012) is the virtual continuous (real-valued) word counts (discussed in Sec. 3.3). Specifically, we define the likelihood in the third term of Eqn. 2 as:

$$p(\mathbf{e}_{mj}|z_{mj}, \boldsymbol{\mu}_{z_{mj}}, \boldsymbol{\Sigma}_{z_{mj}}) = [\mathcal{N}(\mathbf{e}_{mj}; \boldsymbol{\mu}_{mj}, \boldsymbol{\Sigma}_{mj})]^{w_{mj}}. \tag{3}$$

Note that Eqn. 3 is the likelihood of $w_{mj}$ (virtual) words, where $w_{mj}$ can be a continuous value derived from the PLM's attention weights (details in Sec. 3.3).

Correspondingly, in the third item of Eqn. 2, we have:

$$\mathbb{E}_q[\log p(\mathbf{e}_{mj}|z_{mj}, \boldsymbol{\mu}_{z_{mj}}, \boldsymbol{\Sigma}_{z_{mj}})] = \sum_{m,j,k} \phi_{mjk} w_{mj} \log \mathcal{N}(\mathbf{e}_{mj}|\boldsymbol{\mu}_k, \boldsymbol{\Sigma}_k)$$

$$= \sum_{m,j,k} \phi_{mjk} w_{mj} \{-\tfrac{1}{2}(\mathbf{e}_{mj} - \boldsymbol{\mu}_k)^T \boldsymbol{\Sigma}_k^{-1}(\mathbf{e}_{mj} - \boldsymbol{\mu}_k) - \log[(2\pi)^{d/2}|\boldsymbol{\Sigma}_k|^{1/2}]\}. \tag{4}$$

**Update Rules.** Taking the derivative of the ELBO in Eqn. 2 w.r.t. $\phi_{mjk}$ (see Appendix A for details) and setting it to 0 yields the update rule for $\phi_{mjk}$:

$$\phi_{mjk} \propto \frac{w_{mj}}{|\boldsymbol{\Sigma}_k|^{1/2}} \exp[\Psi(\gamma_{mk}) - \Psi(\sum_{k'} \gamma_{mk'}) - \tfrac{1}{2}(\mathbf{e}_{mj} - \boldsymbol{\mu}_k)^T \boldsymbol{\Sigma}_k^{-1}(\mathbf{e}_{mj} - \boldsymbol{\mu}_k)], \tag{5}$$

with the normalization constraint $\sum_{k=1}^K \phi_{mjk} = 1$.

$$\gamma_{mk} = \alpha_k + \sum_{j=1}^{J_m} \phi_{mjk} w_{mj}, \tag{6}$$

where $\boldsymbol{\alpha} \triangleq [\alpha_k]_{k=1}^K$ is the hyperparameter for the Dirichlet prior distribution of $\boldsymbol{\theta}_m$. In summary, the inference algorithm will alternate between updating $\phi_{mjk}$ for all $(m, j, k)$ tuples and updating $\gamma_{mk}$ for all $(m, k)$ tuples.

### 3.4.2 LEARNING

**Learning Dataset-Level Concept Parameters.** The inference algorithm in Sec. 3.4.1 assumes availability of the dataset-level (global) concept parameters $\{(\boldsymbol{\mu}_k, \boldsymbol{\Sigma}_k)\}_{k=1}^K$. To learn such these parameters, one needs to iterate between (1) inferring document-level variational parameters $\boldsymbol{\gamma}_m$ as well as word-level variational parameters $\boldsymbol{\phi}_{mj}$ in Sec. 3.4.1 and (2) learning dataset-level concept parameters $\{(\boldsymbol{\mu}_k, \boldsymbol{\Sigma}_k)\}_{k=1}^K$.

**Update Rules.** Similar to Sec. 3.4.1, we expand the ELBO in Eqn. 2 (see Appendix A for details) and set its derivative w.r.t. $\boldsymbol{\mu}_k$ and $\boldsymbol{\Sigma}_k$ to 0, yielding the update rule for learning $\boldsymbol{\mu}_k$ and $\boldsymbol{\Sigma}_k$:

$$\boldsymbol{\mu}_k = \frac{\sum_{m,j} \phi_{mjk} w_{mj} \mathbf{e}_{mj}}{\sum_{m,j} \phi_{mjk} w_{mj}}, \qquad \boldsymbol{\Sigma}_k = \frac{\sum_{m,j} \phi_{mjk} w_{mj} (\mathbf{e}_{mj} - \boldsymbol{\mu}_k)(\mathbf{e}_{mj} - \boldsymbol{\mu}_k)^T}{\sum_{m,j} \phi_{mjk} w_{mj}}. \tag{7}$$

**Effect of Attention Weights.** From Eqn. 7, we can observe that the attention weight of the $j$'th word in document $m$, i.e., $a_{mj}$, affects the virtual continuous word count $w_{mj}$ (see Sec. 3.3), thereby affecting the update of the dataset-level concept center $\boldsymbol{\mu}_k$ and covariance $\boldsymbol{\Sigma}_k$.

Specifically, if we use attention-based weights with fixed length or variable length in Sec. 3.3, the continuous word count $w_{mj}$ will be proportional to the attention weight $a_{mj}$. Therefore, when updating the concept center $\boldsymbol{\mu}_k$ as a weighted average of different word embeddings $\mathbf{e}_{mj}$, VALANCE naturally places more focus on words with higher attention weights $a_{mj}$ from PLMs, thereby making the interpretations sharper (see Sec. 5.4 for detailed results and Appendix I for theoretical analysis).

---

**Algorithm 1:** Algorithm for VALANCE

**Input:** Initialized $\{\boldsymbol{\gamma}_m\}_{m=1}^M$, $\{\boldsymbol{\phi}_m\}_{m=1}^M$, and $\{\boldsymbol{\Omega}_k\}_{k=1}^K$, documents $\{\mathcal{D}_m\}_{m=1}^M$, number of epochs T.

**for** $t = 1 : T$ **do**
   **for** $m = 1 : M$ **do**
      Update $\boldsymbol{\phi}_m$ and $\boldsymbol{\gamma}_m$ using
   Eqn. 5 and Eqn. 6, respectively.
      Update $\{\boldsymbol{\Omega}_k\}_{k=1}^K$ using Eqn. 7.

---

Interestingly, we also observe that PLMs' attention weights on stop words such as 'the' and 'a' tend to be much lower; therefore VALANCE can naturally ignore these concept-irrelevant stop words when learning and inferring concepts (topics). This is in contrast to typical topic models (Blei et al., 2003; Blei, 2012) that require preprocessing to remove stop words.

## 4 THEORETICAL ANALYSIS

In this section, we provide theoretical guarantees that our VALANCE satisfies the four properties in Definition 3.1.

**Multi-Level Structure.** As shown in Alg. 1, VALANCE (1) learns the *dataset-level* interpretation $\{\boldsymbol{\Omega}_k\}_{k=1}^K$ describing the $K$ concepts, (2) infers the distribution of *document-level* interpretation $\boldsymbol{\theta}_m$ for document $m$, i.e., $q(\boldsymbol{\theta}_m|\boldsymbol{\gamma}_m)$, which is parameterized by $\boldsymbol{\gamma}_m$, and (3) infers the posterior distribution of *word-level* concept index, i.e., $q(z_{mj}|\boldsymbol{\phi}_{mj})$, parameterized by $\boldsymbol{\phi}_{mj}$. Such three-level interpretations correspond to Property (1) in Definition 3.1.

**Normalization.** The variational distribution $q(\boldsymbol{\theta}_m|\boldsymbol{\gamma}_m)$ (Eqn. 1) which VALANCE learns is a Dirichlet distribution; therefore we have $\sum_{k=1}^K \theta_{mk} = 1$. The update of $\phi_{mj}$ (Eqn. 5) is naturally constrained by $\sum_{k=1}^K \phi_{mjk} = 1$ since $\phi_{mj}$ parameterizes a Categorical distribution (over $z_{mj}$).

**Additivity.** VALANCE is able to add or subtract the learned concept activation $\mu_k$ from PLM embeddings via the following Quadratic Programming (QP) problem ($\mathbf{x} = [x_k]_{k=1}^K$):

$$\min_{\mathbf{x} \in \mathbb{R}^K} \quad \| \sum_{k=1}^K x_k \boldsymbol{\mu}_k - \mathbf{e}_m \|^2, \qquad \text{subject to} \quad \mathbf{x} \geq \mathbf{0} \text{ and } \sum_{k=1}^K x_k = 1. \tag{8}$$

Given learned concepts $\{(\boldsymbol{\mu}_k, \boldsymbol{\Sigma}_k)\}_{k=1}^K$, VALANCE obtains this QP's optimal solution $\mathbf{x}^* \in \mathbb{R}^K$ and add/subtract any concept $k$ from arbitrary PLM embedding $\mathbf{e}_m$ by: $\mathbf{e}_m \leftarrow \mathbf{e}_m \pm x_k^* \boldsymbol{\mu}_k$. Alg. 2 summarizes this concept editing process; one can also replace $\mathbf{e}_{mj}$ with the CLS embedding $\mathbf{c}_m$ for document-level editing (see Appendix D for details).

**Mutual Information Maximization.** Theorem 4.1 below shows that our inferred document-level and word-level interpretation, $\boldsymbol{\theta}_m$ and $\{\boldsymbol{\phi}_{mj}\}_{j=1}^{J_m}$, satisfy Property (4), Mutual Information Maximization, in Definition 3.1.

**Theorem 4.1 (Mutual Information Maximization).** *In Eqn. 2, the ELBO* $\mathcal{L}(\gamma_m, \{\phi_{mj}\}_{j=1}^{J_m}; \alpha, \{(\mu_k, \Sigma_k)\}_{k=1}^{K})$ *is upper bounded by the mutual information between contextual embeddings* $\mathbf{e}_m$ *and multi-level interpretation* $\theta_m, \{\phi_{mj}\}_{j=1}^{J_m}$ *in Definition 3.1. Formally, with approximate posteriors* $q(\theta_m|\gamma_m)$ *and* $q(z_{mj}|\phi_{mj})$, *we have*

$$\mathcal{L}(\gamma_m, \{\phi_{mj}\}_{j=1}^{J_m}; \alpha, \{(\mu_k, \Sigma_k)\}_{k=1}^{K}) \leq I(\mathbf{e}_m; \theta_m, \{z_{mj}\}_{j=1}^{J_m}) - H(\mathbf{e}_m), \qquad (9)$$

*where the entropy term* $H(\mathbf{e}_m)$ *is a constant.*

From Theorem 4.1 we can see that maximizing the ELBO in Eqn. 2 is equivalent to maximizing the mutual information between our document-level/word-level concepts and the observed contextual embeddings in PLMs (proof is provided in Appendix H).

In summary, VALANCE enjoys all four properties in Definition 3.1 and therefore generates the optimal conceptual interpretation for PLMs. In contrast, state-of-the-art methods only satisfy a small part of them (Table 2 and Sec. 5.2). In Appendix I, we provide theoretical guarantees that (1) under mild assumptions our VALANCE can learn better conceptual interpretations for PLMs for in noisy data and (2) attention-based schemes is superior to the identical scheme (Sec. 3.3).

## 5 EXPERIMENTS

### 5.1 EXPERIMENT SETUP

**Datasets.** We use three datasets with various text lengths in our experiments, namely 20 Newsgroups, M10 (Lim & Buntine, 2015), and BBC News (Greene & Cunningham, 2006). We follow Terragni et al. (2021) and Zhang et al. (2022) to pre-process these datasets. The statistics of the datasets are summarized in Table 1. We use the standard 8:1:1 train/validation/test set split.

Table 1: Dataset statistics, including the number of documents ($M$), vocabulary size ($V$), the number of corpus categories ($L$), and the average document length ($\overline{J}$).

| Dataset | $M$ | $V$ | $L$ | $\overline{J}$ |
|---|---|---|---|---|
| 20 Newsgroups | 16,309 | 1,612 | 20 | 48 |
| M10 | 8,355 | 1,696 | 10 | 5.9 |
| BBC News | 2,225 | 2,949 | 5 | 120 |

**Baselines.** We compare our method with the following state-of-the-art baselines:

- **SHAP and LIME** Lundberg & Lee (2017); Ribeiro et al. (2016) are interpretation methods that attribute importance scores to input features. In this paper, we use embeddings of 'CLS' token as input to SHAP/LIME.
- **BERTopic** Grootendorst (2020) is a clustering-based model that uses HDBSCAN (McInnes & Healy, 2017) to cluster sentence embeddings from BERT, performs Uniform Manifold Approximation Projection (UMAP) (McInnes et al., 2018), and then uses class-based TF-IDF (c-TF-IDF) to obtain words for each cluster.
- **CETopic** Zhang et al. (2022) is a clustering-based model that first uses UMAP to perform dimensionality reduction on BERT sentence embeddings, performs K-Means clustering (Lloyd, 1982), and then uses weighted word selection for each cluster.

**Evaluation Metric.** Inspired by Koh et al. (2020), we perform concept editing experiments to evaluate conceptual interpretation for PLMs; higher *accuracy gain* after editing indicates better interpretation performance. We leverage BERT-base-uncased (Devlin et al., 2018) as the contextual embedding model. To compare our learned concepts with the baseline models, we first follow their configurations (Grootendorst, 2020; Zhang et al., 2022) to fix BERT model pa-

**Algorithm 2:** Algorithm for VALANCE Concept Editing

---
**Input:** PLM $f(\cdot)$, classifier $g(\cdot)$, classification loss $L$, document $\mathcal{D}_m$ with $J_m$ words, labels $\mathbf{y}$, constant factor $\omega$.
**for** $j = 1 : J_m$ **do** $\quad \mathbf{e}_{mj} = f(\mathcal{D}_{mj})$
$\quad \mathbf{x}^* = QP(\mathbf{e}_{mj}, \{\mu_k\}_{k=1}^{K})$
$\quad k^* = \arg\min_k L(g(\mathbf{e}_{mj} - \omega \cdot x_k^* \mu_k), y_m)$
$\quad \mathbf{e}_{mj} \leftarrow \mathbf{e}_{mj} - \omega \cdot x_{k^*}^* \mu_{k^*}$

---

rameters when learning the topics/concepts, train a classifier on top of the fixed contextual embeddings, and then perform concept pruning (Koh et al., 2020) for different evaluated models on the same classifier. We use accuracy on the test set as our metric.

We can perform concept editing on either input tokens or contextual embeddings of PLMs. Specifically, we can perform *hard* concept editing for concept $k$ by directly removing tokens that belong

concept $k$ (applicable for hard clustering methods such as our baselines); we could also perform *soft* concept editing for concept $k$ by removing concept subspace vectors from contextual embeddings $\mathbf{e}_m$ (applicable for VALANCE using Alg. 2). For SHAP/LIME, we treat the 'CLS' token's embedding as the input features to interpret.

## 5.2 COMPARISON ON FOUR PROPERTIES IN DEFINITION 3.1

In Sec. 4 we show that VALANCE satisfies the four properties of conceptual interpretation in Definition 3.1. In contrast, baseline models do not necessarily learn concepts that meet these requirements. Table 2 summarizes the comparison between our VALANCE and the baselines. We can see that VALANCE is superior to baselines in terms of the following four aspects:

Table 2: Comparing different methods on the four properties in Definition 3.1 (MIM: Mutual Information Maximization).

| Model | Multi-Level | Normalization | Additivity | MIM |
|---|---|---|---|---|
| SHAP/LIME | No | No | Partial | No |
| BERTopic | No | Hard | Partial | No |
| CETopic | No | Hard | Partial | No |
| VALANCE | **Yes** | **Soft** | **Full** | **Yes** |

(1) **Multi-Level Structure.** Baselines either apply clustering algorithms directly on the document-level embeddings from PLMs and therefore can only provide document-level interpretation or assign importance scores to input features, and thus can only provide single-level interpretation. In contrast, VALANCE provides dataset-level, document-level, and word-level interpretation.
(2) **Normalization.** BERTopic and CETopic assign each word to exactly one concept and therefore satisfies *hard*-normalization. SHAP/LIME produce importance scores that are not normalized. In contrast, VALANCE learn fractional concept interpretations $\gamma_m$ and $\phi_{mj}$ and therefore satisfies *soft*-normalization, which is more flexible and intuitive.
(3) **Additivity.** Baselines cannot perform complete addition or subtraction of language concepts, because they operate only at a single level (i.e., word or document). In contrast, VALANCE's additivity and concept editing (Alg. 2) work for both document and word levels.
(4) **Mutual Information Maximization.** Baselines either use a multi-step pipeline or produce only importance scores; they are therefore prone to lose information between raw PLM embeddings and final clustering/scoring results. In contrast, VALANCE is theoretically guaranteed to maximally preserve information (Theorem 4.1).

## 5.3 CONCEPT EDITING RESULTS

**Accuracy Gain.** We perform greedy concept editing (Koh et al., 2020) for BERTopic, CETopic, and our VALANCE to evaluate the quality of their learned concepts. Higher accuracy gain after pruning indicates better performance. Specifically, we perform concept pruning to the CLS embeddings for VALANCE (see details in Alg 2). Since BERTopic and CETopic can infer concepts (topics) only at the document level, their only choice is to prune a concept by completely removing input tokens assigned to the concept (as mentioned in Sec. 5.1 and 5.2).

Table 3: **Accuracy gain on 20 Newsgroups (20NG), M10, and BBC News (BBC) (%).** We mark the best result with **bold face** and the second best results with underline.

| Dataset | | Unedited | SHAP /LIME | BERTopic | CETopic | VALANCE | Finetune (Oracle) |
|---|---|---|---|---|---|---|---|
| 20NG | Acc. | 51.26 | 61.74 | 60.76 | 61.93 | **62.54** | 64.38 |
| | Gain | - | 10.48 | 9.50 | 10.67 | **11.28** | 13.12 |
| M10 | Acc. | 69.74 | 75.60 | 76.79 | 79.18 | **80.74** | 82.54 |
| | Gain | - | 5.86 | 7.05 | 9.44 | **11.00** | 12.80 |
| BBC | Acc. | 93.72 | 95.96 | 95.52 | **96.86** | 96.41 | 97.76 |
| | Gain | - | 2.24 | 1.80 | **3.14** | 2.69 | 4.04 |

Table 3 show the results for different methods in three real-world datasets, where 'Finetune (Oracle)' refers to finetuning both the backbone and the classifier of BERT. VALANCE's concept editing can improve the accuracy upon the unedited model by more than $11\%$ in 20 Newsgroups and M10, almost on par with 'Finetune (Oracle)'. Compared with the baselines, VALANCE achieves the most accuracy gain in 20 Newsgroups and M10 and the second most accuracy gain in BBC News, demonstrating the effectiveness of VALANCE's four properties in Definition 3.1. Note that SHAP and LIME both interpret the CLS token's embedding and therefore has identical accuracy gain (see Appendix E for details).

**Ablation Study.** Thanks to its full additivity (Definition 3.1), VALANCE is capable of different concept editing schemes, including 'Random', 'Unweighted', and 'Weighted'. Specifically, *weighted* pruning uses the concept editing algorithm in Alg. 2 with the optimal hyperparameter $\omega$; *unweighted* pruning runs Alg. 2 with $\omega = 1$; *random* pruning first randomly picks a concept, sets $\omega \cdot x_k = 1/K, \forall k \in \{1, ..., K\}$, and then runs Alg. 2.

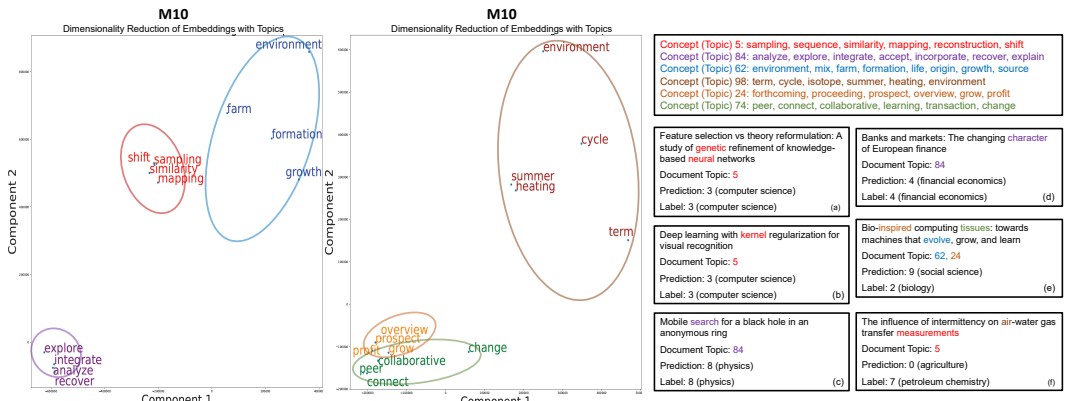

Figure 3: Visualization of VALANCE's three-level conceptual interpretation. **Left and Middle:** Dataset-level interpretation with 6 concepts' $\boldsymbol{\mu}_k$ and $\boldsymbol{\Sigma}_k$ with nearest word embeddings. For better readability, we show 3 concepts in each plot. **Right:** Top words in each concept and 6 example documents with the associated document-level and word-level interpretations.

Table 4 shows accuracy for VALANCE's different schemes. As expected, random pruning barely improves upon the unedited model. Unweighted pruning improves upon the unedited model by $1.5 \sim 3.5\%$. Weighted pruning improves the accuracy by around $11\%$ upon the unedited model on 20 Newsgroups and M10.

Table 4: **VALANCE Editing Accuracy (%).** We mark the best result with **bold face** and the second best results with underline.

| Dataset | Unedited | Random | Unweighted | Weighted | Finetune (Oracle) |
|---|---|---|---|---|---|
| 20 Newsgroups | 51.26 | 51.13 | 54.63 | **62.54** | 64.38 |
| M10 | 69.74 | 69.76 | 73.56 | **80.74** | 82.54 |
| BBC News | 93.72 | 93.72 | 95.52 | **96.41** | 97.76 |

## 5.4 CONCEPTUAL INTERPRETATION (MORE FOR DIFFERENT TASKS IN APPENDIX F)

**Dataset-Level Interpretations.** As a case study, we train VALANCE on M10, sample 6 concepts (topics) from the dataset, and plot the word embeddings of the top words (closest to the center $\boldsymbol{\mu}_k$) in these concepts using PCA in Fig. 3(left and middle). We can observe Concept 5 is mostly about data analysis, including words such as 'sampling' and 'similarity'. Concept 84 is mostly about reasoning, with words 'explore', 'accept', 'explain', etc. Concept 62 is mostly about nature, with words 'environment', 'formation', 'growth', etc. Concept 98 is mostly about farming, with words 'term', 'summer', 'heating', etc. Concept 24 is mostly about economics, with words 'forthcoming', 'prospect', 'grow', etc. Concept 74 is mostly about social contact, containing words such as 'peer', 'connect', and 'collaborative'. Interestingly, Concept 24 (economics) and Concept 74 (social contact) are both related to social science and are therefore closer to each other in Fig. 3(middle), while Concept 98 (farming) is farther away, showing VALANCE's cability of capturing concept similarity.

**Document-Level Interpretations.** Fig. 3(right) shows that VALANCE can provide conceptual interpretations on why correct or incorrect PLM predictions happen for specific documents. For example, document (e) belongs to class 2 (**biology**), but BERT misclassifies it as class 9 (**social science**); our VALANCE interprets that this is because document (e) involves Concept 24 (economics), which is related to **social science**. On the other hand, document (b) is related to machine learning and BERT correctly classifies it as class 3 (**computer science**); VALANCE interprets that this is because document (b) involves Concept 5 (data analysis).

**Word-Level Interpretations.** Fig. 3(right) also shows that VALANCE can interpret which words and what concepts of these words lead to specific PLM predictions. For example, document (f) belongs to class 7 (**petroleum chemistry**), but BERT misclassifies it as class 0 (**agriculture**); VALANCE attributes this to the word 'air', which belongs to Concept 98 (farming). For document (b), VALANCE interprets that BERT correctly classifies it as class 3 (**computer science**) because the document contains the word 'kernel', which belongs to Concept 5 (data analysis).

## 6 CONCLUSION

We identify the problem of multi-level interpretations for PLM predictions, develop a formal definition of conceptual interpretation, and propose VALANCE as the first general method to infer such conceptual interpretation, with promising empirical results. Our theoretical analysis shows that VALANCE is guaranteed to generate the optimal conceptual interpretation by our definition. Future work includes extending VALANCE beyond BERT variants and natural language processing.

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

## A   DETAILS ON LEARNING VALANCE

**Update Rules.** Similar to Sec. 3.4.1 of the main paper, we expand the ELBO in Eqn. 2 of the main paper, take its derivative w.r.t. $\boldsymbol{\mu}_k$ and set it to 0:

$$\frac{\partial L}{\partial \boldsymbol{\mu}_k} = \sum_{m,j} \phi_{mjk} w_{mj} \boldsymbol{\Sigma}_k^{-1} (\mathbf{e}_{mj} - \boldsymbol{\mu}_k) = 0, \tag{10}$$

yielding the update rule for learning $\boldsymbol{\mu}_i$:

$$\boldsymbol{\mu}_k = \frac{\sum_{m,j} \phi_{mjk} w_{mj} \mathbf{e}_{mj}}{\sum_{m,j} \phi_{mjk} w_{mj}}, \tag{11}$$

where $\boldsymbol{\Sigma}_k^{-1}$ is canceled out. Similarly, setting the derivatives w.r.t. $\boldsymbol{\Sigma}$ to 0, i.e.,

$$\frac{\partial L}{\partial \boldsymbol{\Sigma}_k} = \frac{1}{2} \sum_{m,j} \phi_{mjk} w_{mj} (-\boldsymbol{\Sigma}_k^{-1} + \boldsymbol{\Sigma}_k^{-1} (\mathbf{e}_{mj} - \boldsymbol{\mu}_k)(\mathbf{e}_{mj} - \boldsymbol{\mu}_k)^T \boldsymbol{\Sigma}_k^{-1}), \tag{12}$$

we have

$$\boldsymbol{\Sigma}_k = \frac{\sum_{m,j} \phi_{mjk} w_{mj} (\mathbf{e}_{mj} - \boldsymbol{\mu}_k)(\mathbf{e}_{mj} - \boldsymbol{\mu}_k)^T}{\sum_{m,j} \phi_{mjk} w_{mj}}. \tag{13}$$

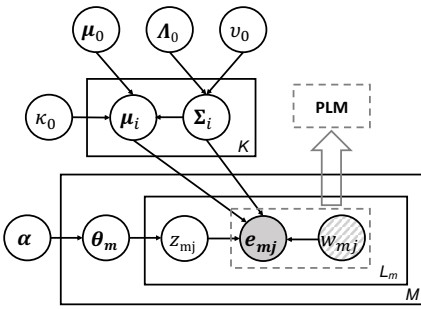

Figure 4: Probabilistic graphical model of smoothed VALANCE.

**Smoothing with Prior Distributions on** $\{(\boldsymbol{\mu}_k, \boldsymbol{\Sigma}_k)\}_{k=1}^K$**.** To alleviate overfitting and prevent singularity in numerical computation, we impose priors distributions on $\boldsymbol{\mu}_k$ and $\boldsymbol{\Sigma}_k$ to smooth the learning process (Fig. 4). Specifically, we use a Normal-Inverse-Wishart prior on $\boldsymbol{\mu}_i$ and $\boldsymbol{\Sigma}_i$:

$$\boldsymbol{\Sigma}_k \sim \mathcal{IW}(\boldsymbol{\Lambda}_0, \nu_0),$$
$$\boldsymbol{\mu}_k | \boldsymbol{\Sigma}_k \sim \mathcal{N}(\boldsymbol{\mu}_0, \boldsymbol{\Sigma}_k/\kappa_0),$$

where $\boldsymbol{\Lambda}_0$, $\nu_0$, $\boldsymbol{\mu}_0$, and $\kappa_0$ are hyperparameters for the prior distributions. Taking the expectations of $\boldsymbol{\mu}_k$ and $\boldsymbol{\Sigma}_k$ over the posterior distibution $\mathcal{NIW}(\boldsymbol{\mu}_k, \boldsymbol{\Sigma}_k | \boldsymbol{\mu}_k^{(n)}, \boldsymbol{\Lambda}_k^{(n)}, \kappa_k^{(n)}, \nu_k^{(n)})$, we have the update rules as:

$$\boldsymbol{\mu}_k \leftarrow \mathbb{E}_{\mathcal{NIW}}[\boldsymbol{\mu}_k] = \frac{\kappa_0 \boldsymbol{\mu}_0 + n_k \widetilde{\boldsymbol{\mu}}_k}{\kappa_0 + n_k}, \tag{14}$$

$$\boldsymbol{\Sigma}_k \leftarrow \mathbb{E}_{\mathcal{NIW}}[\boldsymbol{\Sigma}_k] = \frac{\boldsymbol{\Lambda}_0 + \mathbf{S}_k + \frac{\kappa_0 n_k}{\kappa_0 + n_k}(\widetilde{\boldsymbol{\mu}}_k - \boldsymbol{\mu}_0)(\widetilde{\boldsymbol{\mu}}_k - \boldsymbol{\mu}_0)^T}{\nu_0 + n_k - K - 1}, \tag{15}$$

$$\mathbf{S}_k = \sum\nolimits_{m,j} \phi_{mjk} w_{mj} (\mathbf{e}_{mj} - \widetilde{\boldsymbol{\mu}}_k)(\mathbf{e}_{mj} - \widetilde{\boldsymbol{\mu}}_k)^T. \tag{16}$$

where $n_k = \sum_{m,j} \phi_{mjk} w_{mj}$ is the total virtual word counts used to estimate $\boldsymbol{\mu}_k$ and $\boldsymbol{\Sigma}_k$. Eqn. 14 and Eqn. 15 are the smoothed version of Eqn. 7 of the main paper. From the Bayesian perfective, they correspond to the expectations of $\boldsymbol{\mu}_k$'s and $\boldsymbol{\Sigma}_k$'s posterior distributions. Alg. 1 of the main paper summarizes the learning of VALANCE.

**Online Learning of $\boldsymbol{\mu}_k$ and $\boldsymbol{\Sigma}_k$.** Note that PLMs are deep neural networks trained using minibatches of data, while Eqn. 14 and Eqn. 15 need to go through the whole dataset before each update. Inspired by Hoffman et al. (2010); Oord et al. (2017), we using exponential moving average (EMA) to work with minibatchs. Specifically, we update them as:

$$\boldsymbol{\mu}_k \leftarrow \rho \cdot N \cdot \boldsymbol{\mu}_k + (1 - \rho) \cdot B \cdot \widetilde{\boldsymbol{\mu}}_k,$$
$$\boldsymbol{\Sigma}_k \leftarrow \rho \cdot N \cdot \boldsymbol{\Sigma}_k + (1 - \rho) \cdot B \cdot \widetilde{\boldsymbol{\Sigma}}_k,$$
$$N \leftarrow \rho \cdot N + (1 - \rho) \cdot B,$$
$$\boldsymbol{\mu}_k \leftarrow \frac{\boldsymbol{\mu}_k}{N}, \quad \boldsymbol{\Sigma}_k \leftarrow \frac{\boldsymbol{\Sigma}_k}{N},$$

where $B$ is the minibatch size, $N$ is a running count, and $\rho \in (0, 1)$ is the momentum hyperparameter. $\widetilde{\boldsymbol{\mu}}_k$ and $\widetilde{\boldsymbol{\Sigma}}_k$ are the updated $\boldsymbol{\mu}_k$ and $\boldsymbol{\Sigma}_k$ after applying Eqn. 14 and Eqn. 15 only on the *current minibatch*.

## B    INTERPRETATION OF THE ELBO

We can expand the ELBO in Eqn. 2 of the main paper as:

$$
\begin{aligned}
\mathcal{L}(\boldsymbol{\gamma}, \boldsymbol{\phi}; \boldsymbol{\alpha}, \{\boldsymbol{\mu}\}_{k=1}^K, \{\boldsymbol{\Sigma}\}_{k=1}^K) = {} & \log \boldsymbol{\Gamma}(\sum_{k=1}^K \alpha_k) - \sum_{k=1}^K \log \boldsymbol{\Gamma}(\alpha_k) + \sum_{k=1}^K (\alpha_k - 1)(\Psi(\boldsymbol{\gamma}_k) - \Psi(\sum_{k'=1}^K \boldsymbol{\gamma}_{k'})) \\
& + \sum_{j=1}^J \sum_{k=1}^K \phi_{jk}(\Psi(\boldsymbol{\gamma}_k) - \Psi(\sum_{k'=1}^K \boldsymbol{\gamma}_{k'})) \\
& + \sum_{j,k} \phi_{jk} w_j \{-\tfrac{1}{2}(\mathbf{e}_j - \boldsymbol{\mu}_k)^T \boldsymbol{\Sigma}_k^{-1}(\mathbf{e}_j - \boldsymbol{\mu}_k) - \log[(2\pi)^{d/2}|\boldsymbol{\Sigma}_k|^{1/2}]\} \\
& - \log \boldsymbol{\Gamma}(\sum_{k=1}^K \boldsymbol{\gamma}_j) + \sum_{k=1}^K \log \boldsymbol{\Gamma}(\boldsymbol{\gamma}_k) - \sum_{k=1}^K (\boldsymbol{\gamma}_k - 1)(\Psi(\boldsymbol{\gamma}_k) - \Psi(\sum_{k'=1}^K \boldsymbol{\gamma}_{k'})) \\
& - \sum_{j=1}^J \sum_{k=1}^K \phi_{jk} \log \phi_{jk}.
\end{aligned} \tag{17}
$$

We can interpret the meaning of each term of ELBO as follows:

- The sum of the first and the fourth terms, namely $\mathbb{E}_q[\log p(\boldsymbol{\theta}_m | \boldsymbol{\alpha})] - \mathbb{E}_q[\log q(\boldsymbol{\theta}_m)]$, is equal to $-KL(q(\boldsymbol{\theta}_m)|p(\boldsymbol{\theta}_m|\boldsymbol{\alpha}))$, which is the negation of KL Divergence between the variational posterior probability $q(\boldsymbol{\theta}_m)$ and the prior probability $p(\boldsymbol{\theta}_m|\alpha)$ of the topic proportion $\boldsymbol{\theta}_m$ for document $m$. Therefore maximizing the sum of these two terms is equivalent to minimizing the KL Divergence $KL(q(\boldsymbol{\theta}_m)|p(\boldsymbol{\theta}_m|\boldsymbol{\alpha}))$; this serves as a regularization term to make sure the inferred $q(\boldsymbol{\theta}_m)$ is close to its prior distribution $p(\boldsymbol{\theta}_m|\boldsymbol{\alpha})$.

- Similarly, the sum of the second and the last terms (ignoring the summation over the word index $j$ for simplicity), namely $\mathbb{E}_q[\log p(z_{mj}|\boldsymbol{\theta}_m)] - \mathbb{E}_q[\log q(z_{mj})]$ is equal to $-KL(q(z_{mj})|p(z_{mj}|\boldsymbol{\theta}_m))$, which is the negation of the KL Divergence between the variational posterior probability $q(z_{mj})$ and the prior probability $p(z_{mj}|\boldsymbol{\theta}_m)$ of the word-level topic assignment $z_{mj}$ for word $j$ of document $m$. Therefore maximizing the sum of these two terms is equivalent to minimizing the KL Divergence $KL(q(z_{mj})|p(z_{mj}|\boldsymbol{\theta}_m))$; this serves as a regularization term to make sure the inferred $q(z_{mj})$ is close to its "prior" distribution $p(z_{mj}|\boldsymbol{\theta}_m)$.

- The third term $\sum_{j=1}^{J_m} \mathbb{E}_q[\log p(\mathbf{e}_{mj}|z_{mj}, \boldsymbol{\mu}_{z_{mj}}, \boldsymbol{\Sigma}_{z_{mj}})]$ is to maximize the log likelihood $p(\mathbf{e}_{mj}|z_{mj}, \boldsymbol{\mu}_{z_{mj}}, \boldsymbol{\Sigma}_{z_{mj}})$ of every contextual embedding $\mathbf{e}_{mj}$ (for word $j$ of document $m$) conditioned on the inferred $z_{mj}$ and the parameters $(\boldsymbol{\mu}_{z_{mj}}, \boldsymbol{\Sigma}_{z_{mj}})$.

## C  EXPERIMENTAL SETTINGS AND IMPLEMENTATION DETAILS

We will release all code, models, and data. Below we provide more details on the experimental settings and practical implementation.

**Datasets.** We use the GLUE benchmark (Wang et al., 2018) to perform *additional* conceptual interpretation in this section. This benchmark includes multiple sub-tasks of predictions, with the paired sentences as inputs. In this paper, we use 4 datasets from GLUE (MRPC, RTE, STS-B, and QQP) to show contextual interpretations.

**Visualization Postprocessing.** For better showcase the dataset-level concepts as in Fig. 3 of the main paper, we may employ simple linear transformations on the embedding of words after the aforementioned PCA step, in order to scatter all the informative words on the same figures. However, for some datasets such as STS-B, this is not necessary; therefore we do not use it for these datasets.

**Topic (Concept) Identification.** Inspired by Blei et al. (2003), we identify meaningful topics by listing the top-5 topics for each word, computing the inverse document frequency (IDF), and filtering out topics with the lowest IDF scores. Note that although GLUE benchmark are datasets that consists of documents with small size, making it particularly challenging for traditional topic models (such as LDA) to learn topics; interestingly our VALANCE can still perform well in learning the topics. We contribute this to the following observations: (1) Compared to traditional LDA using *discrete* word representations, VALANCE uses *continuous* word embeddings. In such a continuous space, topics learned for one word can also help neighboring words; this alleviates the sparsity issue caused by short documents and therefore learns better topics. (2) VALANCE's attention-based continuous word counts further improves sample efficiency. In VALANCE, important words have larger attention weights and therefore larger continuous word counts. In this case, *one* important word in a sentence possesses statistical (sample) power equivalent to *multiple* words; this leads to better sample efficiency in VALANCE.

---

**Algorithm 3:** Algorithm for VALANCE Document-Level Concept Editing

**Input:** PLM $f(\cdot)$, classifier $g(\cdot)$, classification loss $L$, dataset $\{\mathcal{D}_m\}_{m=1}^M$, labels $\mathbf{y}$, constant factor $\omega$.

**for** $m = 1 : M$ **do**
  $\mathbf{c}_m = f(\mathcal{D}_m)$
  $\mathbf{x}^* = QP(\mathbf{c}_m, \{\boldsymbol{\mu}_k\}_{k=1}^K)$
  $k^* = \arg\min_{k=1}^K L(g(\mathbf{c}_m - \omega \cdot x_k^* \boldsymbol{\mu}_k), y_m)$
  $\mathbf{c}_m \leftarrow \mathbf{c}_m - \omega \cdot x_{k^*}^* \boldsymbol{\mu}_{k^*}$

---

## D  DOCUMENT-LEVEL CONCEPT EDITING

We describe the document-level concept eding algorithm of VALANCE in Alg. 3. $c_m$ denotes the 'CLS' embedding of document $m$ (see Fig. 1 of the main paper).

## E  MORE DETAILS ON CONCEPT EDITING

Assume each BERT model contains a backbone and a classifier. To perform concept editing:

(1) We first train a classifier on top of the *fixed* BERT embeddings generated by the *fixed* backbone to get the original accuracy in the "Unedited" column (in Table 3 and Table 4 of the main paper).

(2) We then apply the same embedding cluster methods to these BERT embeddings to infer the concepts/topics for each dataset.

(3) Finally, with the inferred concepts/topics from the baselines (SHAP/LIME, BERTopic and CETopic in Table 3 of the main paper) and our VALANCE variants (Unweighted and Weighted in Table 4 of the main paper), we perform concept editing and feed the concept-edited embeddings into the trained classifier from Step (1) to compute the editing accuracy for different methods.

Since here one *does not fully finetune the BERT model* (i.e., keeping the backbone fixed), the editing accuracy is expected to be lower than the "Finetune" column (in Table 3 and Table 4 of the main paper), which serves as the oracle. Table Table 3 of the main paper shows that our VALANCE learns better concepts than the baselines, and Table 4 of the main paper shows that the weighted variant of VALANCE performs better.

Note that SHAP and LIME both interpret the CLS token's embedding, and hence their concept vectors have the same dimension as the PLM embedding vector (768 in our case). When we conduct concept editing on the $k$'th dimension/concept, we simply subtract the CLS embedding's dimension $k$ with the average value in the batch on dimension $k$ (which means that we know little about the concept/dimension $k$ on this document), and keep values of the other dimensions unchanged. Note that the pruning process is exactly the same for SHAP and LIME. Therefore SHAP and LIME have identical test accuracy and accuracy gain.

## F    MORE CONCEPTUAL INTERPRETATION RESULTS IN DIFFERENT DOWNSTREAM TASKS

**Dataset-Level Interpretations.**    As in the main paper, we leverage VALANCE as an interpreter on MRPC, RTE, STS-B and QQP, respectively, sample $3, 3, 4, 4$ concepts (topics) for each dataset respectively, and plot the word embeddings of the top words (closest to the center $\boldsymbol{\mu}_i$) in these concepts using PCA. Fig. 5(left) shows the concepts from MRPC. We can observe Concept 20 is mostly about policing, including words such as 'suspect', 'police', and 'house'. Concept 24 is mostly about politics, including words such as 'capital', 'Congress', and 'Senate'. Concept 27 contains mostly names such as 'Margaret' and 'Mary'. Similarly, Fig. 5(right) shows the concepts from RTE. We can observe Concept 67 is related to West Asia and includes words such as 'Quran' and 'Pasha'. Concept 13 is related to Europe and includes European countries/names such as 'Prussia' and 'Salzburg'. Concept 91 is mostly about healthcare and includes words such as 'physiology' and 'insulin'. Fig. 6 shows the concepts from STS-B. We can observe Concept 63 is mostly about household and daily life, including words such as 'trash', 'flowers', 'airs', and 'garden'. Concept 60 is mostly about tools, including words such as 'stations', 'rope', 'parachute', and 'hose'. Concept 84 is mostly about national security, including words such as 'guerilla', 'NSA', 'espionage', and 'raided'. Concept 55 contains mostly countries and cities such as 'Kiev', 'Moscow', 'Algeria', and 'Ukrainian'. Similarly, Fig. 7 shows the concepts from QQP. We can observe that Concept 12 is mostly about negative attitude, including words such as 'boring', 'criticism', and 'blame'. Concept 73 is mostly about Psychology, including words such as 'adrenaline', 'haunting', and 'paranoia'. Concept 34 is mostly about prevention and conservatives, including words such as 'destroys', 'unacceptable', and 'prohibits'. Concept 64 is mostly about strategies, including words such as 'rumours', 'boycott', and 'deportation'.

**Document-Level Interpretations.** For document-level conceptual interpretations, we sample two example documents from MRPC (Fig. 5(left)), three from RTE (Fig. 5(right)), six from STS-B (Fig. 6) and eight from QQP (Fig. 7), respectively, where each document contains a pair of sentences. The MRPC task is to predict whether one sentence paraphrases the other. For example, in the first document of MRPC, we can see that our VALANCE correctly interprets the model prediction 'True' with Concept 24 (politics). The RTE task is to predict whether one sentence entail the other. For example, in the second document of RTE, VALANCE correctly interprets the model prediction 'True' with Concept 13 (countries). The STS-B task is to predict the semantic similarity between

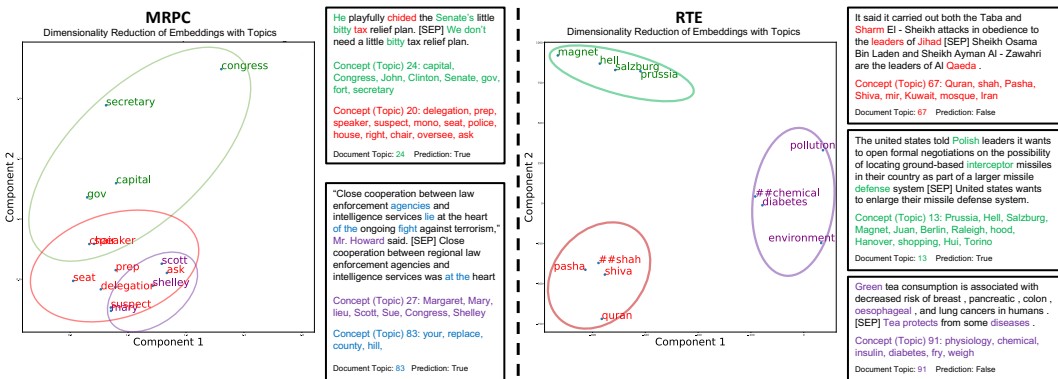

Figure 5: Visualization of VALANCE's learned topics of contextual word embeddings. **Left:** MRPC's dataset-level interpretation with two example documents. Concept 83 is relatively far from the other three concepts in the embedding space; therefore we omit it on the left panel for better readability. **Right:** RTE's dataset-level interpretation with three example documents.

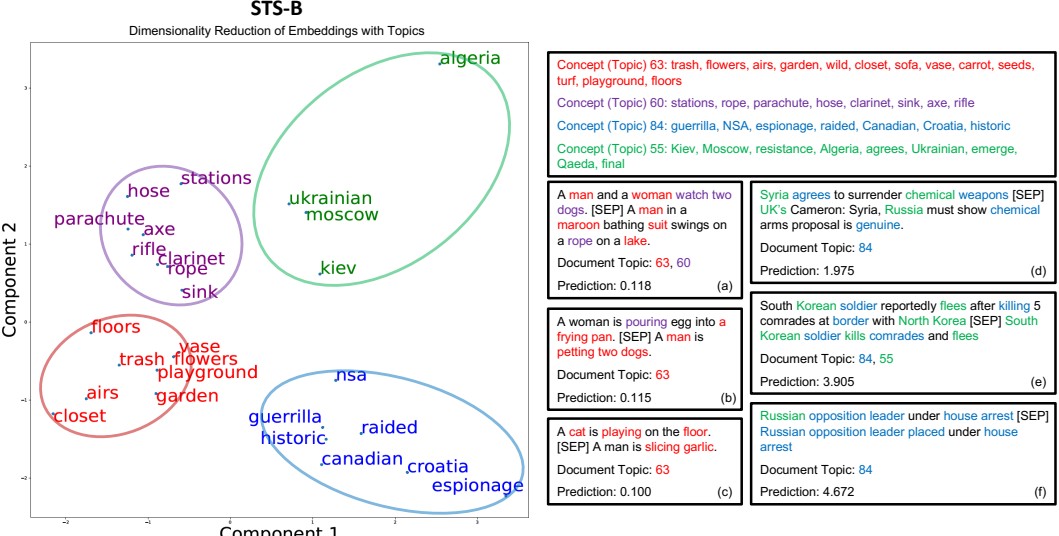

Figure 6: Visualization of VALANCE's learned topics of contextual word embeddings. We show STS-B's dataset-level interpretation with six example documents. The prediction of VALANCE is between the range of $[0, 5]$.

two sentences with the score range of $[0, 5]$. For example, in Document (a) of Fig. 6, we can see that VALANCE correctly interpret the model's predicted similarity score '$0.118$' (which is relatively low,) with Concept 63 (household and daily life) and Concept 60 (tools). Similarly, in Document (f) of Fig. 6, we can see that VALANCE correctly interpret the model's predicted similarity score '$4.672$' (which is relatively high) with Concept 84 (national security). The QQP task is to predict whether the two questions are paraphrase of each other. For example, in Document (b) of Fig. 7, we can see that VALANCE correctly interprets the model's predicted label 'False' with Concept 73 (Psychology). Similarly, in Document (e) of Fig. 7, we can see that VALANCE correctly interprets the model's predicted label 'True' with Concept 64 (strategies).

**Word-Level Interpretations.** For word-level conceptual interpretations, we can observe that VALANCE interpret the PLM's prediction on MRPC's first document (Fig. 5(left)) using words such as 'senate' and 'bitty' that are related to politics. Note that the word 'bitty' is commonly used (with 'little') by politicians to refer to the small size of tax relief/cut plans. Similarly, for RTE's first document (Fig. 5(right)), VALANCE correctly identifies Concept 67 (West Asia) and interprets the model prediction 'False' by distinguishing between keywords such as 'Jihad' and 'Al

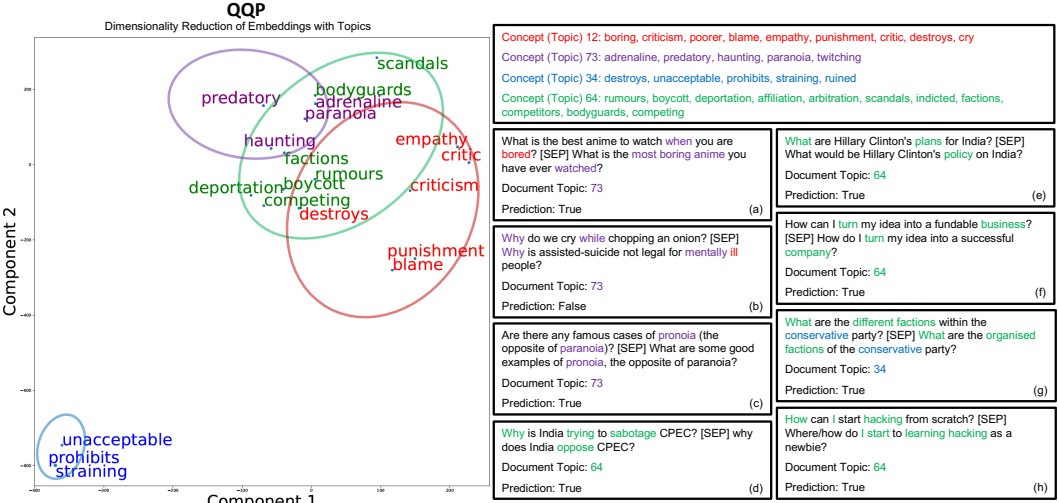

Figure 7: Visualization of VALANCE's learned topics of contextual word embeddings. We show QQP's dataset-level interpretation with eight example documents.

Table 5: Example concepts on RTE dataset learned by VALANCE.

| Concepts | Top Words | | | | | | | |
|---|---|---|---|---|---|---|---|---|
| **bio-chem** | cigarette | biological | ozone | cardiovascular | chemist | liver | chemical | toxin |
| **citizenship** | indies | bolivian | fiji | surrey | jamaican | dutch | latino | caribbean |
| **names** | mozart | spielberg | einstein | bush | kurt | liszt | hilton | lynn |
| **conspiracy** | secretly | corrupt | disperse | infected | ill | hidden | illegally | sniper |
| **administration** | reagan | interior | ambassador | prosecutor | diplomat | legislative | spokesman | embassy |
| **crime** | fraud | laundering | sheriff | prosecutor | corruption | fool | robber | greed |

Qaeda'. likewise, we can observe that VALANCE interprets PLM's prediction on Document (c) of Fig. 6 using words such as 'cat', 'floor', and 'garlic' that are related to household and daily life. Also, VALANCE interprets PLM's prediction on Document (e) of Fig. 6 using words such as 'soldier' and 'border' that are related to national security. Similarly, for QQP's Document (d) (Fig. 7), VALANCE correctly interprets the model prediction 'True' by identifying keywords such as 'sabotage' and 'oppose' with similar meanings in the topic of strategies. For QQP's Document (g), (Fig. 7), VALANCE interprets the words in the both sentences with the same semantics, such as 'conservative' that is related to prevention and conservatives (note that in politics, 'conservative' refers to parties that tend to prevent/block new policies or legislation), and thereby predicting the correct label 'True'.

**Example Concepts.** Following Blei et al. (2003), we show the learned concepts on the RTE dataset in Table 5, which is complementary to aforementioned explanations. We select several different topics from Fig. 5. As in Sec. 5.4 of the main paper, we obtain top words from each concept via first calculating the average of the each word's corresponding contextual embeddings over the dataset, and then getting the nearest words to each topic center ($\mu_k$) in the embedding space. As we can see in Table 5, VALANCE can capture various concepts with profound and accurate semantics. Therefore, although PLM embeddings are contextual and continuous, our VALANCE can still find conceptual patterns of words on the dataset-level.

# G  DOCUMENT CLASSIFICATION WITH VALANCE CONCEPTS

We ran additional experiments to perform document classification with the 'CLS' token's embedding and $\theta$ (inferred from VALANCE) as features. Table 6 below shows the results on three datasets. The results show that our VLANCE can learn meaningful concept vector $\theta$, which can improve model predictions of document labels.

Table 6: Comparison of Unedited and Unedited+$\boldsymbol{\theta}$ on 20 Newsgroups, M10, and BBC News

|  | Unedited | Unedited+$\boldsymbol{\theta}$ |
|---|---|---|
| 20 Newsgroups | 51.26 | **51.74** |
| M10 | 69.74 | **70.76** |
| BBC News | 93.72 | **94.90** |

## H  THEORY ON THE MUTUAL INFORMATION MAXIMIZATION PROPERTY

We provide the following proof of Theorem 4.1 of the main paper.

For convenience, let $\Omega = (\boldsymbol{\mu}_{k=1}^{K}, \Sigma_{k=1}^{K})$, and $\beta = (\boldsymbol{\theta}_m, \mathbf{z}_m)$.

We then introduce a helper joint distribution of the variables $\mathbf{e}_m$ and $\beta$, $s(\mathbf{e}_m, \beta) = p(\mathbf{e}_m)q(\beta|\mathbf{e}_m)$.

According to the definition of ELBO of Section 3.4.1, in Eqn. 9, we have

$$LHS = \mathcal{L}(\gamma_m, \phi_m; \alpha, \Omega) = \mathbb{E}_{p(\mathbf{e}_m)}[\mathbb{E}_{q(\beta)}[\log p(\mathbf{e}_m|\Omega, \beta)]] + \mathbb{E}_{q(\beta)}[\log q(\beta|\Omega)]. \quad (18)$$

Since $\mathbb{E}_{q(\beta)}[\log q(\beta|\Omega)] \leq 0$, we only need to prove that

$$\mathbb{E}_{p(\mathbf{e}_m)}[\mathbb{E}_{q(\beta)}[\log p(\mathbf{e}_m|\Omega, \beta)]] \leq I_s(\mathbf{e}_m; \beta) - H(\mathbf{e}_m) = RHS. \quad (19)$$

Then we have that

$$
\begin{aligned}
\mathbb{E}_{p(\mathbf{e}_m)}[\mathbb{E}_q[\log p(\mathbf{e}_m|\beta, \Omega)]] &\leq \mathbb{E}_{p(\mathbf{e}_m)}[\mathbb{E}_q[\log p(\mathbf{e}_m|\beta)]] \\
&= \mathbb{E}_{p(\mathbf{e}_m)}[\mathbb{E}_q[\log \tfrac{q(\mathbf{e}_m|\beta)}{p(\mathbf{e}_m)} \tfrac{p(\mathbf{e}_m)p(\mathbf{e}_m|\beta)}{q(\mathbf{e}_m|\beta)}]] \\
&= \mathbb{E}_{p(\mathbf{e}_m)}[\mathbb{E}_q[\log \tfrac{q(\mathbf{e}_m|\beta)}{p(\mathbf{e}_m)}]] + \mathbb{E}_{p(\mathbf{e}_m)}[\mathbb{E}_q[\log p(\mathbf{e}_m)]] + \mathbb{E}_{p(\mathbf{e}_m)}[\mathbb{E}_q[\log \tfrac{p(\mathbf{e}_m|\beta)}{q(\mathbf{e}_m|\beta)}]] \\
&= I_s(\mathbf{e}_m; \beta) - H(\mathbf{e}_m) - \mathbb{E}_q[KL(q(\mathbf{e}_m|\beta)|p(\mathbf{e}_m|\beta))] \\
&\leq I_s(\mathbf{e}_m; \beta) - H(\mathbf{e}_m) - 0 = RHS,
\end{aligned}
\quad (20)
$$

which concludes the proof of Theorem 4.1.

## I  THEORETICAL ANALYSIS ON CONTINUOUS WORD COUNTS

Before going to the claims and proofs, first we specify some basic problem settings and assumptions. Suppose there are $K+1$ topic groups, each of which is regarded to be sampled from a parameterized multivariate Gaussian distribution. In specific, the $K+1$'th distribution of topic has a much larger covariance, and in the same time, closed to the center of embedding space. The prementioned properties can be measured by a series of inequalities:

The approximate marginal log-likelihood of word embeddings, i.e., the third term of the ELBO as mentioned in Eqn. 2 of the main paper, is:

$$
\begin{aligned}
\mathcal{L}^{(train)} &= \sum_{j=1}^{J_m} \mathbb{E}_q[\log p(\mathbf{e}_{mj}|z_{mj}, \boldsymbol{\mu}_{z_{mj}}, \boldsymbol{\Sigma}_{z_{mj}})] \\
&= \sum_{m,j,k} \phi_{mjk} w_{mj}\{-\tfrac{1}{2}(\mathbf{e}_{mj} - \boldsymbol{\mu}_k)^T \boldsymbol{\Sigma}_k^{-1}(\mathbf{e}_{mj} - \boldsymbol{\mu}_k) - \log[(2\pi)^{d/2}|\boldsymbol{\Sigma}_k|^{1/2}]\}.
\end{aligned}
\quad (21)
$$

The above equation is the training objective, yet for fair comparison of different training schemes, we calculate the approximated likelihood with word count 1 for all words.

$$
\begin{aligned}
\mathcal{L}^{(eval)} &= \sum_{j=1}^{J_m} \mathbb{E}_q[\log p'(\mathbf{e}_{mj}|z_{mj}, \boldsymbol{\mu}_{z_{mj}}, \boldsymbol{\Sigma}_{z_{mj}})] \\
&= \sum_{m,j,k} \phi_{mjk}\{-\tfrac{1}{2}(\mathbf{e}_{mj} - \boldsymbol{\mu}_k)^T \boldsymbol{\Sigma}_k^{-1}(\mathbf{e}_{mj} - \boldsymbol{\mu}_k) - \log[(2\pi)^{d/2}|\boldsymbol{\Sigma}_k|^{1/2}]\}.
\end{aligned}
\quad (22)
$$

### I.1 GAUSSIAN MIXTURE MODELS

Suppose we have a ground truth GMM model with parameters $\boldsymbol{\pi}^* \in \mathbb{R}^K$ and $\{\boldsymbol{\mu}_k^*, \boldsymbol{\Sigma}_k^*\}_{k=1}^K$, with $K$ different Gaussian distributions. In the dataset, let $N$ and $N_s$ denote the numbers of non-stop-words and stop-words, respectively. Then the marginal log likelihood of a learned GMM model on a given data sample $\mathbf{e}$ can be written as

$$p(\mathbf{e}|\{\boldsymbol{\mu}, \boldsymbol{\Sigma}\}, \boldsymbol{\pi}) = \sum_{k=1}^K \boldsymbol{\pi}_k \mathcal{N}(\mathbf{e}; \boldsymbol{\mu}_k, \boldsymbol{\Sigma}_k). \tag{23}$$

Assuming a dataset of $N + N_s$ words $\{\mathbf{e}_i\}_{i=1}^{N+N_s}$ and taking the associated weights $w_i$ for each word into account, the log-likelihood of the dataset can be written as

$$\sum_{i=1}^{N+N_s} p(\mathbf{e}_i|\{\boldsymbol{\mu}_k, \boldsymbol{\Sigma}_k\}_{k=1}^K, \boldsymbol{\pi}) = \sum_{i=1}^N \log \sum_{k=1}^K w_i \boldsymbol{\pi}_k \mathcal{N}(\mathbf{e}_i; \boldsymbol{\mu}_k, \boldsymbol{\Sigma}_k) + \sum_{i=N+1}^{N+N_s} \log \sum_{k=1}^K w_i \boldsymbol{\pi}_k \mathcal{N}(\mathbf{e}_i; \boldsymbol{\mu}_k, \boldsymbol{\Sigma}_k). \tag{24}$$

Leveraging Jensen's inequality, we obtain a lower bound of the above quantity (denoting as $\boldsymbol{\Theta}$ the collection of parameters $\{\boldsymbol{\mu}_k, \boldsymbol{\Sigma}_k\}_{k=1}^K$ and $\boldsymbol{\pi}$):

$$\mathcal{L}_{\text{GMM}}(\boldsymbol{\Theta}, \{w_i\}) = \sum_{i=1}^N w_i \log \sum_{k=1}^K \pi_k \mathcal{N}(\mathbf{e}_i; \boldsymbol{\mu}_k, \boldsymbol{\Sigma}_k) + \sum_{i=N+1}^{N+N_s} w_i \log \sum_{k=1}^K \pi_k \mathcal{N}(\mathbf{e}_i; \boldsymbol{\mu}_k, \boldsymbol{\Sigma}_k) + C, \tag{25}$$

where C is a constant.

In the following theoretical analysis, we consider the following three different configurations of the weights $w_i$.

**Definition I.1** (**Weight Configurations**). We define three different weight configurations as follows:

- Identical Weights: $w_i = \frac{1}{N+N_s}, i \in \{1, 2, \ldots, N + N_s\}$

- Ground-Truth Weights : $w_i = \begin{cases} \frac{1}{N}, & i \in \{1, 2, \ldots, N\} \\ 0, & i \in \{N+1, N+2, \ldots, N+N_s\} \end{cases}$

- Attention-Based Weights: $w_i = \begin{cases} \lambda_1 \in [\frac{1}{N+N_s}, \frac{1}{N}], & i \in \{1, 2, \ldots, N\} \\ \lambda_2 \in [0, \frac{1}{N+N_s}], & i \in \{N+1, N+2, \ldots, N+N_s\} \end{cases}$

**Definition I.2** (**Advanced Weight Configurations**). We define three different weight configurations as follows:

- Identical Weights: $w_i = \frac{1}{N+N_s}, i \in \{1, 2, \ldots, N + N_s\}$

- Ground-Truth Weights : $w_i = \begin{cases} \frac{1}{N}, & i \in \{1, 2, \ldots, N\} \\ 0, & i \in \{N+1, N+2, \ldots, N+N_s\} \end{cases}$

- Attention-Based Weights: $w_i \in \begin{cases} [\frac{1}{N+N_s}, \frac{1}{N}], & i \in \{1, 2, \ldots, N\} \\ [0, \frac{1}{N+N_s}], & i \in \{N+1, N+2, \ldots, N+N_s\} \end{cases}$

**Definition I.3** (**Optimal Parameters**). With Definition I.1, the corresponding optimal parameters are then defined as follows:

$$\boldsymbol{\Theta}_I = \arg \max_{\boldsymbol{\Theta}} \mathcal{L}(\boldsymbol{\Theta}; \mathbf{w} \to \text{Identical}), \tag{26}$$

$$\boldsymbol{\Theta}_G = \arg \max_{\boldsymbol{\Theta}} \mathcal{L}(\boldsymbol{\Theta}; \mathbf{w} \to \text{GT}), \tag{27}$$

$$\boldsymbol{\Theta}_A = \arg \max_{\boldsymbol{\Theta}} \mathcal{L}(\boldsymbol{\Theta}; \mathbf{w} \to \text{Attention}), \tag{28}$$

where $\mathbf{w} \to \text{Identical}$, $\mathbf{w} \to \text{GT}$, and $\mathbf{w} \to \text{Attention}$ indicates that 'Identical Weights', 'Ground-Truth Weights', and 'Attention-Based Weights' are used, respectively.

**Lemma I.1.** *Suppose we have two series of functions $\{f_{1,i}(x)\}$ and $\{f_{2,i}(x)\}$, with two non-negative weighting parameters $\lambda_1, \lambda_2$ satisfying $N\lambda_1 + N_s\lambda_2 = 1$. We define the final objective function $f(\cdot)$ as:*

$$f(x; \lambda_1, \lambda_2) = \lambda_1 \sum_{i=1}^{N} f_{1,i}(x) + \lambda_2 \sum_{i=N+1}^{N_s} f_{2,i}(x). \tag{29}$$

*We assume two pairs of parameters $(\lambda_1, \lambda_2)$ and $(\lambda_1', \lambda_2')$, where*

$$\lambda_1 \geq \lambda_1', \tag{30}$$
$$\lambda_2 \leq \lambda_2'. \tag{31}$$

*Defining the optimal values of the objective function for different weighting parameters as*

$$\widehat{x} = \arg\max_x f(x; \lambda_1, \lambda_2), \tag{32}$$
$$\widehat{x}' = \arg\max_x f(x; \lambda_1', \lambda_2'), \tag{33}$$

*we then have that*

$$f(\widehat{x}; \tfrac{1}{N}, 0) \geq f(\widehat{x}'; \tfrac{1}{N}, 0). \tag{34}$$

*Proof.* We prove this theorem by contradiction. Suppose that we have

$$f(\widehat{x}; \tfrac{1}{N}, 0) < f(\widehat{x}'; \tfrac{1}{N}, 0). \tag{35}$$

According to Eqn. 30, i.e., $\lambda_1 \geq \lambda_1'$, and the equation $N\lambda_1 + N_s\lambda_2 = 1$, we have

$$\lambda_1\lambda_2' = \lambda_1 \frac{1 - N\lambda_1'}{N_s} \geq \lambda_1' \frac{1 - N\lambda_1}{N_s} = \lambda_1'\lambda_2. \tag{36}$$

According to Eqn. 33, we have the following equality:

$$f(\widehat{x}; \lambda_1', \lambda_2') \leq f(\widehat{x}'; \lambda_1', \lambda_2'). \tag{37}$$

Combined with the aforementioned assumption in Eqn. 35, we have that

$$\lambda_2' f(\widehat{x}; \lambda_1, \lambda_2) = \lambda_1\lambda_2' \sum_{i=1}^{N} f_{1,i}(\widehat{x}) + \lambda_2\lambda_2' \sum_{i=N+1}^{N_s} f_{2,i}(\widehat{x}) \tag{38}$$

$$= (\lambda_1'\lambda_2 \sum_{i=1}^{N} f_{1,i}(\widehat{x}) + \lambda_2'\lambda_2 \sum_{i=N+1}^{N_s} f_{2,i}(\widehat{x})) + (N(\lambda_1\lambda_2' - \lambda_1'\lambda_2) \cdot \tfrac{1}{N} \sum_{i=1}^{N} f_{1,i}(\widehat{x})) \tag{39}$$

$$= \lambda_2 f(\widehat{x}; \lambda_1', \lambda_2') + N(\lambda_1\lambda_2' - \lambda_1'\lambda_2) f(\widehat{x}; \tfrac{1}{N}, 0) \tag{40}$$

$$< \lambda_2 f(\widehat{x}'; \lambda_1', \lambda_2') + N(\lambda_1\lambda_2' - \lambda_1'\lambda_2) f(\widehat{x}'; \tfrac{1}{N}, 0) \tag{41}$$

$$= (\lambda_1'\lambda_2 \sum_{i=1}^{N} f_{1,i}(\widehat{x}') + \lambda_2'\lambda_2 \sum_{i=N+1}^{N_s} f_{2,i}(\widehat{x}')) + (N(\lambda_1\lambda_2' - \lambda_1'\lambda_2) \cdot \tfrac{1}{N} \sum_{i=1}^{N} f_{1,i}(\widehat{x}')) \tag{42}$$

$$= \lambda_1\lambda_2' \sum_{i=1}^{N} f_{1,i}(\widehat{x}') + \lambda_2\lambda_2' \sum_{i=N+1}^{N_s} f_{2,i}(\widehat{x}') \tag{43}$$

$$= \lambda_2' f(\widehat{x}'; \lambda_1, \lambda_2), \tag{44}$$

which contradicts the definition of $\widehat{x}$ in Eqn. 32 (i.e., $\widehat{x}$ maximizes $f(x; \lambda_1, \lambda_2)$), completing the proof. □

**Lemma I.2.** *Suppose we have two series of functions $\{f_{1,i}(x)\}$ and $\{f_{2,i}(x)\}$, with two series of non-negative weighting parameters $\boldsymbol{\lambda}_1 = [\lambda_{1,i}]_{i=1}^{N}, \boldsymbol{\lambda}_2 = [\lambda_{2,i}]_{i=N+1}^{N_s}$ satisfying $\sum_{i=1}^{N} \lambda_{1,i} + \sum_{i=N+1}^{N_s} \lambda_{2,i} = 1$. We define the final objective function $f(\cdot)$ as:*

$$f(x; \boldsymbol{\lambda}_1, \boldsymbol{\lambda}_2) = \sum_{i=1}^{N} \lambda_{1,i} f_{1,i}(x) + \sum_{i=N+1}^{N_s} \lambda_{2,i} f_{2,i}(x). \tag{45}$$

We assume two pairs of parameters $(\boldsymbol{\lambda}_1, \boldsymbol{\lambda}_2)$ and $(\boldsymbol{\lambda}_1', \boldsymbol{\lambda}_2')$, where

$$\lambda_{1,i} \geq \lambda_{1,i}', \quad i \in \{1, 2, ..., N\}, \tag{46}$$

$$\lambda_{2,i} \leq \lambda_{2,i}', \quad i \in \{N+1, N+2, ..., N_s\}. \tag{47}$$

Defining the optimal values of the objective function for different weighting parameters as

$$\widehat{x} = \arg\max_x f(x; \boldsymbol{\lambda}_1, \boldsymbol{\lambda}_2), \tag{48}$$

$$\widehat{x}' = \arg\max_x f(x; \boldsymbol{\lambda}_1', \boldsymbol{\lambda}_2'), \tag{49}$$

$$x^* = \arg\max f(x, \tfrac{1}{N}, \mathbf{0}). \tag{50}$$

Under the following **Assumptions** (with $\mathbf{1}$ and $\mathbf{0}$ denoting vectors with all entries equal to $1$ and $0$, respectively):

1. $f(\widehat{x}, \mathbf{0}, \boldsymbol{\lambda}_2) \leq f(\widehat{x}', \mathbf{0}, \boldsymbol{\lambda}_2)$.

2. $f(x; \boldsymbol{\lambda}, \mathbf{0}) \geq f(x'; \boldsymbol{\lambda}, \mathbf{0})$, iff $\|x - x^*\| \leq \|x' - x^*\|$, $\boldsymbol{\lambda} \geq 0$, $\|\boldsymbol{\lambda}\|_1 = 1$.

we have that

$$f(\widehat{x}; \tfrac{1}{N}, \mathbf{0}) \geq f(\widehat{x}'; \tfrac{1}{N}, \mathbf{0}). \tag{51}$$

*Proof.* We start with proving the following equality by contradiction:

$$\|\widehat{x} - x^*\| \leq \|\widehat{x}' - x^*\|. \tag{52}$$

Specifically, if

$$\|\widehat{x} - x^*\| > \|\widehat{x}' - x^*\|, \tag{53}$$

leveraging the Assumption 1 and 2 above, we have that

$$f(\widehat{x}; \boldsymbol{\lambda}_1, \boldsymbol{\lambda}_2) = f(\widehat{x}; \boldsymbol{\lambda}_1, \mathbf{0}) + f(\widehat{x}; \mathbf{0}, \boldsymbol{\lambda}_2) < f(\widehat{x}'; \boldsymbol{\lambda}_1, \mathbf{0}) + f(\widehat{x}'; \mathbf{0}, \boldsymbol{\lambda}_2) = f(\widehat{x}'; \boldsymbol{\lambda}_1, \boldsymbol{\lambda}_2), \tag{54}$$

which contradicts Eqn. 48. Therefore, Eqn. 52 holds.

Combining Eqn. 52 and Assumption 2 above, we have that

$$f(\widehat{x}; \tfrac{1}{N}, \mathbf{0}) \geq f(\widehat{x}'; \tfrac{1}{N}, \mathbf{0}), \tag{55}$$

concluding the proof. $\qquad\square$

Based on the definitions and lemmas above, we have the following theorems:

**Theorem I.3** (**Advantage of $\boldsymbol{\Theta}_A$ in the Simplified Case**). *With Definition I.1 and Definition I.3, comparing $\boldsymbol{\Theta}_I$, $\boldsymbol{\Theta}_G$, and $\boldsymbol{\Theta}_A$ by evaluating them on the marginal log-likelihood of non-stop-words, i.e., $\mathcal{L}(\cdot, w \to GT)$, we have that*

$$\mathcal{L}_{GMM}(\boldsymbol{\Theta}_I; \mathbf{w} \to GT) \leq \mathcal{L}_{GMM}(\boldsymbol{\Theta}_A; \mathbf{w} \to GT) \leq \mathcal{L}_{GMM}(\boldsymbol{\Theta}_G; \mathbf{w} \to GT). \tag{56}$$

*Proof.* First, by definition one can easily find that $\boldsymbol{\Theta}_G$ achieves the largest $\mathcal{L}(\cdot; \mathbf{w} \to \mathrm{GT})$ among the three:

$$\max[\mathcal{L}_{GMM}(\boldsymbol{\Theta}_I; \mathbf{w} \to \mathrm{GT}), \mathcal{L}_{GMM}(\boldsymbol{\Theta}_A; \mathbf{w} \to \mathrm{GT})] \leq \max_{\boldsymbol{\Theta}} \mathcal{L}_{GMM}(\boldsymbol{\Theta}; \mathbf{w} \to \mathrm{GT}) = \mathcal{L}_{GMM}(\boldsymbol{\Theta}_G; \mathbf{w} \to \mathrm{GT}). \tag{57}$$

Next, we set $\{w_i\}_{i=1}^N$ to $\lambda_1$ and $\{w_i\}_{i=N+1}^{N+N_s}$ to $\lambda_2$, respectively; we rewrite $\log \sum_{k=1}^K \pi_k \mathcal{N}(\mathbf{e}_i; \boldsymbol{\mu}_k, \boldsymbol{\Sigma}_k)$ as $f_{1,i}(x)$ for $i \in \{1, 2, \ldots, N\}$ and $f_{2,i}(x)$ for $i \in \{N+1, N+1, \ldots, N+N_s\}$, where $x$ corresponds to $\boldsymbol{\Theta} \triangleq (\boldsymbol{\pi}, \{\boldsymbol{\mu}_k, \boldsymbol{\Sigma}_k\}_{k=1}^K)$. By Lemma I.1, we have that

$$\mathcal{L}_{\mathrm{GMM}}(\boldsymbol{\Theta}_A; \mathbf{w} \to \mathrm{GT}) \leq \mathcal{L}_{\mathrm{GMM}}(\boldsymbol{\Theta}_G; \mathbf{w} \to \mathrm{GT}). \tag{58}$$

Combining Eqn. 57 and Eqn. 58 concludes the proof. $\qquad\square$

Theorem I.3 shows that under mild assumptions, the attention-based weights can help produce better estimates of $\Theta$ in the presence of noisy stop-words and therefore learns higher-quality topics from the corpus, improving interpretability of PLMs.

**Theorem I.4 (Advantage of $\Theta_A$ in the General Case).** *With Definition I.2 and Definition I.3, comparing $\Theta_I$, $\Theta_G$, and $\Theta_A$ by evaluating them on the marginal log-likelihood of non-stop-words, i.e., $\mathcal{L}_{GMM}(\cdot, w \to GT)$, we have that*

$$\mathcal{L}_{GMM}(\Theta_I; \mathbf{w} \to GT) \leq \mathcal{L}_{GMM}(\Theta_A; \mathbf{w} \to GT) \leq \mathcal{L}_{GMM}(\Theta_G; \mathbf{w} \to GT). \tag{59}$$

*Proof.* First, by definition one can easily find that $\Theta_G$ achieves the largest $\mathcal{L}(\cdot; \mathbf{w} \to GT)$ among the three:

$$\max[\mathcal{L}_{\text{GMM}}(\Theta_I; \mathbf{w} \to GT), \mathcal{L}_{\text{GMM}}(\Theta_A; \mathbf{w} \to GT)] \leq \max_{\Theta} \mathcal{L}_{\text{GMM}}(\Theta; \mathbf{w} \to GT) = \mathcal{L}_{\text{GMM}}(\Theta_G; \mathbf{w} \to GT). \tag{60}$$

Next, we invoke Lemma I.2 by (1) setting $\{w_i\}_{i=1}^N$ to $\boldsymbol{\lambda}_1$ and $\{w_i\}_{i=N+1}^{N+N_s}$ to $\boldsymbol{\lambda}_2$, respectively, and (2) rewriting $\log \sum_{k=1}^K \pi_k \mathcal{N}(\mathbf{e}_i; \boldsymbol{\mu}_k, \boldsymbol{\Sigma}_k)$ as $f_{1,i}(x)$ for $i \in \{1, 2, \ldots, N\}$ and $f_{2,i}(x)$ for $i \in \{N+1, N+1, \ldots, N+N_s\}$, where $x$ corresponds to $\Theta \triangleq (\boldsymbol{\pi}, \{\boldsymbol{\mu}_k, \boldsymbol{\Sigma}_k\}_{k=1}^K)$. By Lemma I.2, we then have that

$$\mathcal{L}_{\text{GMM}}(\Theta_A; \mathbf{w} \to GT) \leq \mathcal{L}_{\text{GMM}}(\Theta_G; \mathbf{w} \to GT). \tag{61}$$

Note that because $f_{1,i}(\cdot)$ and $f_{2,i}(\cdot)$ are Gaussian, therefore Assumption 1 and 2 in Lemma I.2 hold naturally under mild regularity conditions.

Combining Eqn. 60 and Eqn. 61 concludes the proof. $\square$

## I.2 VALANCE AS INTERPRETERS

As mentioned in Eqn. B , the ELBO of the marginal likelihood (denoting as $\Theta$ the collection of parameters $\phi, \gamma$ and $\{\boldsymbol{\mu}_k, \boldsymbol{\Sigma}_k\}_{k=1}^K$) is as follows:

$$\begin{aligned}
\mathcal{L}_{\text{VALANCE}}(\Theta; \{w_i\}) &= \sum_{j=1}^{L'} \mathbb{E}_q[\log p(\mathbf{e}_{mj}|z_{mj}, \boldsymbol{\mu}_{z_{mj}}, \boldsymbol{\Sigma}_{z_{mj}})] \\
&= \sum_{m,j} w_{mj} \sum_k \phi_{mjk} \{-\tfrac{1}{2}(\mathbf{e}_{mj} - \boldsymbol{\mu}_k)^T \boldsymbol{\Sigma}_k^{-1}(\mathbf{e}_{mj} - \boldsymbol{\mu}_k) - \log[(2\pi)^{H/2}|\boldsymbol{\Sigma}_k|^{1/2}]\}.
\end{aligned} \tag{62}$$

Based on the definitions and lemmas above, we have the following theorems:

**Theorem I.5 (Advantage of $\Theta_A$ in the Simplified Case).** *With Definition I.1 and Definition I.3, comparing $\Theta_I$, $\Theta_G$, and $\Theta_A$ by evaluating them on the marginal log-likelihood of non-stop-words, i.e., $\mathcal{L}(\cdot, w \to GT)$, we have that*

$$\mathcal{L}_{VALANCE}(\Theta_I; \mathbf{w} \to GT) \leq \mathcal{L}_{VALANCE}(\Theta_A; \mathbf{w} \to GT) \leq \mathcal{L}_{VALANCE}(\Theta_G; \mathbf{w} \to GT). \tag{63}$$

*Proof.* First, by definition one can easily find that $\Theta_G$ achieves the largest $\mathcal{L}(\cdot; \mathbf{w} \to GT)$ among the three:

$$\max[\mathcal{L}_{\text{VALANCE}}(\Theta_I; \mathbf{w} \to GT), \mathcal{L}_{\text{VALANCE}}(\Theta_A; \mathbf{w} \to GT)] \leq \max_{\Theta} \mathcal{L}_{\text{VALANCE}}(\Theta; \mathbf{w} \to GT) = \mathcal{L}_{\text{VALANCE}}(\Theta_G; \mathbf{w} \to GT). \tag{64}$$

Next, we set $\cup_m \{w_{mj}\}_{j=1}^{N_m}$ to $\boldsymbol{\lambda}_1$ and $\cup_m \{w_{mj}\}_{j=N_m+1}^{N_m+N_{m,s}}$ to $\boldsymbol{\lambda}_2$, respectively; we rewrite $\sum_i \phi_{mji} \{-\tfrac{1}{2}(\mathbf{e}_{mj} - \boldsymbol{\mu}_i)^T \boldsymbol{\Sigma}_i^{-1}(\mathbf{e}_{mj} - \boldsymbol{\mu}_i) - \log[(2\pi)^{d/2}|\boldsymbol{\Sigma}_i|^{1/2}]\}$ as $f_{1,j}(x)$ for $j \in \cup_m \{1, 2, \ldots, N_m\}$ and $f_{2,j}(x)$ for $j \in \cup_m \{N_m + 1, N_m + 1, \ldots, N_m + N_{m,s}\}$, where $x$ corresponds to $\Theta \triangleq (\phi, \gamma, \{\boldsymbol{\mu}_k, \boldsymbol{\Sigma}_k\}_{k=1}^K)$. By Lemma I.1, we have that

$$\mathcal{L}_{\text{VALANCE}}(\Theta_A; \mathbf{w} \to GT) \leq \mathcal{L}_{\text{VALANCE}}(\Theta_G; \mathbf{w} \to GT). \tag{65}$$

Combining Eqn. 64 and Eqn. 65 concludes the proof. $\square$

Theorem I.5 shows that under mild assumptions, the attention-based weights can help produce better estimates of $\Theta$ in the presence of noisy stop-words and therefore learns higher-quality topics from the corpus, improving and interpretability of PLMs.

**Theorem I.6** (**Advantage of $\Theta_A$ in the General Case**). *With Definition I.2 and Definition I.3, comparing $\Theta_I$, $\Theta_G$, and $\Theta_A$ by evaluating them on the marginal log-likelihood of non-stop-words, i.e., $\mathcal{L}_{VALANCE}(\cdot, w \to GT)$, we have that*

$$\mathcal{L}_{VALANCE}(\Theta_I; \mathbf{w} \to GT) \leq \mathcal{L}_{VALANCE}(\Theta_A; \mathbf{w} \to GT) \leq \mathcal{L}_{VALANCE}(\Theta_G; \mathbf{w} \to GT). \qquad (66)$$

*Proof.* First, by definition one can easily find that $\Theta_G$ achieves the largest $\mathcal{L}(\cdot; \mathbf{w} \to \mathrm{GT})$ among the three:

$$\max[\mathcal{L}_{\text{VALANCE}}(\Theta_I; \mathbf{w} \to \mathrm{GT}), \mathcal{L}_{\text{VALANCE}}(\Theta_A; \mathbf{w} \to \mathrm{GT})] \leq \max_{\Theta} \mathcal{L}_{\text{VALANCE}}(\Theta; \mathbf{w} \to \mathrm{GT}) = \mathcal{L}_{\text{VALANCE}}(\Theta_G; \mathbf{w} \to \mathrm{GT}). \quad (67)$$

Next, we invoke Lemma I.2 by (1) setting $\cup_m \{w_{mj}\}_{j=1}^{N_m}$ to $\boldsymbol{\lambda}_1$ and $\cup_m \{w_{mj}\}_{j=N_m+1}^{N_m+N_{m,s}}$ to $\boldsymbol{\lambda}_2$, respectively, and (2) rewriting $\sum_i \phi_{mji}\{-\frac{1}{2}(\mathbf{e}_{mj} - \boldsymbol{\mu}_i)^T \boldsymbol{\Sigma}_i^{-1}(\mathbf{e}_{mj} - \boldsymbol{\mu}_i) - \log[(2\pi)^{d/2}|\boldsymbol{\Sigma}_i|^{1/2}]\}$ as $f_{1,j}(x)$ for $j \in \cup_m \{1, 2, \ldots, N_m\}$ and $f_{2,j}(x)$ for $j \in \cup_m \{N_m + 1, N_m + 1, \ldots, N_m + N_{m,s}\}$, where $x$ corresponds to $\Theta \triangleq (\boldsymbol{\phi}, \boldsymbol{\gamma}, \{\boldsymbol{\mu}_k, \boldsymbol{\Sigma}_k\}_{k=1}^K)$. By Lemma I.2, we then have that

$$\mathcal{L}_{\text{VALANCE}}(\Theta_A; \mathbf{w} \to \mathrm{GT}) \leq \mathcal{L}_{\text{VALANCE}}(\Theta_G; \mathbf{w} \to \mathrm{GT}). \qquad (68)$$

Note that because $f_{1,j}(\cdot)$ and $f_{2,j}(\cdot)$ are very close to Gaussian, therefore Assumption 1 and 2 in Lemma I.2 hold naturally under mild regularity conditions.

Combining Eqn. 67 and Eqn. 68 concludes the proof. $\qquad \square$

