# OpenReview forum: "Variational Language Concepts for Interpreting Pretrained Language Models"
_ICLR.cc/2024/Conference — ICLR 2024 Conference Withdrawn Submission_

### Official Review · Reviewer_Spgz · 2023-10-31

**Soundness:** 2 fair
**Presentation:** 2 fair
**Contribution:** 2 fair
**Rating:** 3
**Confidence:** 3

**Summary:**

This paper proposes a variational approach to detect word-document-topic association from bidirectional attentions in pretrained language models. The proposal is intuitive, and experiments on some benchmark seem strong. The work could be much more impactful if adapted on autoregressive models to support the need for interpretability.

**Strengths:**

The idea of formulating the word-document-topic association in terms of variational lower bound is interesting. Might contribute to previous line of work.

Experiment results on editing accuracy seems particularly strong while the accuracy gain in Tab 3 seems on the marginal side. Overall, it suggests the method has merit over prior works.

**Weaknesses:**

Writing is very confusing and weak. There are many claims in the abstract, intro, and sec 3.1-3.3 I found not convincing.
- In Abs, "lacking in readability and intuitiveness". I think word-level structure is already very readable and intuitive. The thing is, this paper does not say what is the higher-level structure it enables over many prior works, and what is the more readability/intuitiveness it offers. Let alone to say, there are already linguistic patterns found over bidirectional attentions which is already beyond word-level stuff.
- In Intro, "compatible with any attention-based PLMs". I am not convinced on the decoder-only LLMs unless there is experiment to show.
- In Sec 3.2, the definition seems loose. Not sure why the additivity is a bonus. Algebraically, yes it something nice to have. But in what real-world use case?
- In Sec. 3.3 the algo tells little. Writing is confusing.

Verbalizer can be an important baseline but is not presented. There might also be a drawback on the inference speed in this paper, but not reported. The need to train on the ELBo seems another overhead.

Experiments do not come with significance tests.

**Questions:**

Please see the main body of comments.

---

### Official Review · Reviewer_XoeF · 2023-11-01

**Soundness:** 3 good
**Presentation:** 2 fair
**Contribution:** 2 fair
**Rating:** 5
**Confidence:** 3

**Summary:**

The authors propose VAriational LANguage ConcEpt (VALANCE) for interpreting predictions from pretrained language models (PLMs) like BERT at the concept level. The key idea is to model the contextual word embeddings and attention weights from PLMs as observations to infer dataset-level, document-level, and word-level concepts using variational bayesian. The authors gives a theoretical analysis showing VALANCE finds the optimal concepts. The experiments demonstrate VALANCE can successfully provide conceptual interpretations of PLMs at multiple levels.

**Strengths:**

- The authors address an important problem in the field of PLMs: making PLMs interpretable at high levels. They advocate that concept-level interpretation can be a better choice than word-level interpretation offered by attention weights. And the proposed method VALANCE directly addresses them.
- The authors provide a formal definition of conceptual interpretation and proves theorems showing VALANCE satisfies the desired properties.
- The empirical experiments demonstrate the proposed method's strong performances.

**Weaknesses:**

- The paper can be improved by better grounding in the literature.
    - The authors are talking about the pretrain-finetune paradigm for PLMs. However, recent wave of progress on PLMs highlight the paradigm pretrain-prompt. I will suggest expanding the related work in Sec 2 to include it. For example, https://arxiv.org/abs/2005.14165
    - The line of PLM interpretability research is not limited to attention. Probing, mechanistic, natural language interpretation are among the latest ones. The authors should consider mentioning these work and expand Sec 2 accordingly. https://transformer-circuits.pub/2021/framework/index.html https://aclanthology.org/D19-1448/

**Questions:**

- Does the method work with other pretrained models? e.g. GPT-2

---

### Official Review · Reviewer_5MHt · 2023-11-04

**Soundness:** 2 fair
**Presentation:** 2 fair
**Contribution:** 2 fair
**Rating:** 3
**Confidence:** 4

**Summary:**

This paper proposes a concept/topic model over documents where the observed variables are BERT representations for each token and the attention weights of each token embedding to produce [CLS] representation instead of the actual word identities as is usually the case with topic models. The approach and technical details are heavily borrowed from Blei et al. (2002)’s work on Latent Dirichlet Allocation for topic modeling via variational inference. The only difference between the two models is that instead of modeling the density p(token | topic) as a multinomial distribution, this paper treats is as a Gaussian distribution with topic-specific mean and co-variance parameters which is further adjusted by the attention weight contribution of the token to the [CLS] token. This approach is compared to other contextualized/neural topic models like BERTopic, CETopic, and SHAP and LIME interpretability methods on different document classification datasets. The metric used to evaluate is based on concept editing (Koh et al. 2020) which focuses on editing representations to remove information from the embeddings about the most irrelevant/distracting topic in order to improve classification accuracy.

**Strengths:**

– The motivation and the core idea of freezing the BERT model and maximizing the likelihood of the representations of documents under BERT model via a LDA topic model is reasonable and promising.

– The results indicate that the proposed approach seems to outperform other interpretability methods in terms of context editing accuracy gains. However, CETopic seems to be very competitive with the proposed approach even beating it on one of the datasets.

– The qualitative analysis shows a potential use case of the topic model for debugging text classification models.

**Weaknesses:**

– The results are not convincing. As mentioned above, the approach barely beats CETopic and is outperformed by the baseline on one of the datasets. Moreover, I checked multiple times but couldn’t find the criteria for the number of conceps/topics that are fit to the data. In topic modeling literature, this is one of the most important hyperparameters and can impact the results drastically. Since I couldn’t find this information, I am not sure if the comparison to baselines was done fairly. How sensitive is the current approach to the number of topics hyperparameters? Does it beat the baselines over all reasonable ways of setting this hyperparameter?

– The analysis should be more substantial. The only quantitative metric is “accuracy gain under concept editing”. No human studies are performed which is a surprising omission considering that this is a paper focusing on interpretability. Other intrinsic metrics for measuring topical coherence and relevance would also be useful. Moreover, extrinsic metrics focusing on performance on a downstream task using the inferred topic and document representations would also add to the understanding of the impact of the proposed approach.

– The overall writing and presentation needs to be improved significantly – especially the mathematical exposition. The equations and symbols are inconsistent which makes the paper difficult to read. For example, eqn 4 is clearly incorrect: it sums over “m” and “j” variables even though the LHS fixes the “m” and “j” variables. While this is a benign example, I think such a way of presentation also results in deeper issues like the “soft counts” using the attention values seem to cause problems with the model definition (next point).

– As mentioned above, I don’t think the interpretation of attention as soft counts is correctly handled in the paper. For example, equation 3 is defining a “density function” but it is called “likelihood” in the paper. I am not sure if this exponentiated normal distribution is a valid density function. But I can see how if the same embeddings repeated multiple times, one would get the exponentiation effect in the likelihood term. But unclear presentation and such definitional inconsistency has left me unsure about the correctness of the soft counts (w_mj).

– Even if we assume equation 3 to be using attention weights as soft counts (<1) that contribute to the likelihood, the update rules in equations 5 and 6 seem incorrect. Although the update rules derivation is deferred to Appendix A, I couldn’t find the derivations of 5 and 6. When I tried to do it on my own, I think that w_mj should be in the exponent in eqn 5 (instead of linear). And I don’t know how w_mj shows up in eqn 6. Eqn 6 differentiates the objective in eqn 2 wrt parameter \gamma. But the terms containing \gamma in equation 2 don’t seem to involve w_mj at all, hence my confusion about eqn 6. I might be missing something but an elaborate derivation would be helpful.

**Questions:**

Please address the concerns in the weakness section.